# SARS-CoV-2 Reinfection Is a New Challenge for the Effectiveness of Global Vaccination Campaign: A Systematic Review of Cases Reported in Literature

**DOI:** 10.3390/ijerph182011001

**Published:** 2021-10-19

**Authors:** Lorenzo Lo Muzio, Mariateresa Ambosino, Eleonora Lo Muzio, Mir Faeq Ali Quadri

**Affiliations:** 1Department of Clinical and Experimental Medicine, University of Foggia, 70122 Foggia, Italy; mariateresaambrosino@libero.it; 2Consorzio Interuniversitario Nazionale per la Bio-Oncologia (C.I.N.B.O.), 66100 Chieti, Italy; 3Department of Translational Medicine and for Romagna, University of Ferrara, 44121 Ferrara, Italy; eleonoralomuzio@gmail.com; 4Department of Preventive Dental Sciences, Jazan University, Jazan 82511, Saudi Arabia; dr.faeq.quadri@gmail.com

**Keywords:** coronavirus, reinfection, COVID-19, SARS-CoV-2, systematic review

## Abstract

Reinfection with SARS-CoV-2 seems to be a rare phenomenon. The objective of this study is to carry out a systematic search of literature on the SARS-CoV-2 reinfection in order to understand the success of the global vaccine campaigns. A systematic search was performed. Inclusion criteria included a positive RT-PCR test of more than 90 days after the initial test and the confirmed recovery or a positive RT-PCR test of more than 45 days after the initial test that is accompanied by compatible symptoms or epidemiological exposure, naturally after the confirmed recovery. Only 117 articles were included in the final review with 260 confirmed cases. The severity of the reinfection episode was more severe in 92/260 (35.3%) with death only in 14 cases. The observation that many reinfection cases were less severe than initial cases is interesting because it may suggest partial protection from disease. Another interesting line of data is the detection of different clades or lineages by genome sequencing between initial infection and reinfection in 52/260 cases (20%). The findings are useful and contribute towards the role of vaccination in response to the COVID-19 infections. Due to the reinfection cases with SARS-CoV-2, it is evident that the level of immunity is not 100% for all individuals. These data highlight how it is necessary to continue to observe all the prescriptions recently indicated in the literature in order to avoid new contagion for all people after healing from COVID-19 or becoming asymptomatic positive.

## 1. Introduction

The novel coronavirus (SARS-CoV-2) outbreak since December 2019 has continued to exhibit devastating consequences, and was declared as a pandemic by the World Health Organization in early 2020 [1,2,3]. To date, as of 17 October 2021, 240,421,359 infections have been confirmed, with 4,895,034 deaths [4]. In many countries, the vaccination campaign has started with the use of various vaccines recently put on the market and the total number of vaccine doses administered is 6,609,632,994. However, a new problem is emerging with regard to the evolution of the behavior of SARS-CoV-2: the possibility of reinfection of healed subjects after the first infection. On 25 August 2020, the first case of reinfection of SARS-CoV-2 was reported in international literature [5]. This event pointed out that infection by this virus does not uniformly confer protective immunity to all infected individuals [6]. Therefore, several critical questions are intriguing the researchers. Is SARS-CoV-2 reinfection a widespread phenomenon or is it limited to few subjects with immune deficits or specific comorbidities [6]? Can this phenomenon be due to a too weak, too short, or too narrow natural immune response to SARS-CoV-2, that is unable to protect from subsequent exposure [6]? What is the clinical behavior, in regard to the evolution of the reinfections? Can these reinfected patients transmit the viruses? This important problem needs to be addressed, because the possibility of reinfection could drastically reduce the effectiveness of the vaccination campaigns in progress. Protective, sustainable and long-lasting immunity following COVID-19 infection is uncertain, but it is essential for the efficacity of vaccine strategy.

For some viruses, the first infection can provide lifelong immunity, for seasonal coronaviruses protective immunity is short-lived [7]. Over the years, other viruses responsible for various infectious respiratory diseases have been able to present reinfection in the originally cured subjects, such as the coronavirus HCoV-NL63 (NL63) [8] and the human respiratory syncytial virus (hRSV) [9].

The SARS-CoV-2 pandemic poses a challenge regarding the follow-up of recovered patients and the question of the reinfection risk. Several reports confirmed that most patients with SARS-CoV-2 produce antibodies against spike and N-proteins of the virus within 30 days after the infection [10,11]. In fact, an outbreak of the virus on a fishery vessel showed that fishers with prior neutralizing antibodies against SARS-CoV-2 were not reinfected [12]. The potential mechanisms that mediate immunity post-COVID-19 are not yet fully understood. COVID-19 typically follows a course similar to other respiratory viral illnesses, and it is self-limiting in more than 80% of cases [13]. An innate immune response involving T cells and B cells is activated, leading to the production of neutralizing antiviral antibodies [13]. The specific IgM antibody response starts to peak within the first 7 days [13]. Specific IgG and IgA antibodies develop a few days after IgM and are hypothesized to persist at low levels, conferring lifelong protective antibodies [14]. While this hypothesis may hold true for symptomatic patients, emerging data have revealed negative IgM and IgG during the early convalescent phase in asymptomatic patients [15] and 40% of asymptomatic patients became seronegative for IgG 8 weeks after discharging compared with 12.9% who were seronegative for the symptomatic group [15]. A seronegative status could leave open the possibility of reinfection. Immunosuppression and comorbid diseases can be other risk factors for a reinfection [16].

However, a distinction must be made between prolonged shedding/reactivation and true reinfection [17], in fact one of the features of SARS-CoV-2 infection is prolonged virus shedding. Several studies reported persistent or recurrent elimination of viral RNA in nasopharyngeal samples starting from first contact with a positive subject [18,19,20]. For this reason, recently the Center for Disease Control and Prevention (CDC) released a guidance protocol designed to identify cases of real SARS-CoV-2 reinfection [21]. This guidance defines some criteria about sequencing parameters, epidemiological data and laboratory diagnostic data (Table 1). Specifically, investigative criteria include a positive RT-PCR test more than 90 days after the initial test in healed patients or a positive RT-PCR test more than 45 days after the initial test that is accompanied by compatible symptoms or epidemiological exposure, after confirmed healing.

Another emerging problem that can influence the possibility of reinfection and the vaccination efficacity is the new variants of SARS-CoV-2, such us alpha, beta, gamma and delta. A recent study on 9119 patients with SAS-CoV-2 infection identified reinfection in 63 cases (0.7%, 95% confidence interval 0.5–0.9%) [22]. The mean period between two positive tests was 116 ± 21 days [22]. There were no significant differences based on age or sex, while nicotine dependence/tobacco use, asthma were higher in patients with reinfection [22]. There was a significantly lower rate of pneumonia, heart failure, and acute kidney injury during reinfection compared with primary infection [22]. There were two deaths (3.2%) associated with reinfection [22].

Another study conducted in Switzerland reported five cases of reinfection (1%) in 498 seropositive individuals followed for 35 weeks [23]. Breathnach et al. examined data of 10,727 patients with COVID-19 in the first wave and individuated eight reinfection cases (0.07%), all in female patients, and only one was admitted in hospital [24]. Bongiovanni et al. examined 677 subjects with at least a positive nasopharyngeal swab, 328 during the first wave and 349 during the second individuating 13 (1.9%) cases of reinfection [25]. Vitale et al. examined a cohort of 1579 patients and reported five reinfections (0.31%, 95% CI, 0.03–0.58%), of whom only one was hospitalized and the mean (SD) interval between primary infection and reinfection was longer than 230 (90) days [26].

The understanding of COVID-19 reinfection will be key in guiding government and public health policy decisions in the coming months.

A systematic review of literature was performed in order to individuate cases of reinfection for SARS-CoV-2. To date there are more than 300 reported cases of COVID-19 reinfection from different countries such as United States [27], Ecuador [28], Hong Kong [5], and Belgium [29]. It is necessary to understand if all these cases are really reinfection.

## 2. Materials and Methods

This systematic review of literature on reinfections of SARS-CoV-2 was conducted in August 2021. Our study adhered to the Preferred Reporting Items for Systematic Reviews and Meta-Analyses (PRISMA) checklist to ensure the reliability and validity of this study and results.

### 2.1. Data Sources

By application of a systematic search and using the keywords in the online databases including PubMed, Scopus, Web of Science, Science Direct, EMBASE, and preprint servers (MedRxiv, BioRxiv, and SSRN) on 31 July 2021, we extracted all the papers published in English from December 2019 to July 2021. We included several combinations of keywords in the following orders to conduct the search strategy: (1) “CoVID-19” or “SARS-CoV-2” or “2019-nCoV” [all field]; (2) “Reinfection” or “Re-infection” [all field].

### 2.2. Study Selection

Three independent investigators retrieved the studies that were the most relevant by titles and abstracts (ELM, LLM, MA). Subsequently, the full text of the retrieved papers was reviewed, and the most relevant papers were chosen according to the eligibility criteria. Then, we extracted the relevant data and organized them in tables. The original papers that were peer-reviewed and published in English and fulfilled the eligibility criteria were included in the final report, together with two works not reviewed at the time of preparation of this report [30,31].

The following inclusion criteria was used: a positive RT-PCR test carried out more than 90 days after the initial test in healed patients or a positive RT-PCR test carried out more than 45 days after the initial test that is accompanied by compatible symptoms or epidemiological exposure, after confirmed healing. This criteria corresponds to the CDC protocol designed to identify cases of real SARS-CoV-2 reinfection (Table 1) [32].

We considered the exclusion criteria for this study as follows: (1) papers conveying non-human studies including in vitro observations or articles focusing on animal experiments; (2) papers in which their full text were out of access; (3) any suspicious and duplicated results in the databases.

### 2.3. Data Extraction

After summarizing, we transferred the information of the authors, type of article (e.g., case reports), publication date, country of origin, age, gender, and clinical symptoms to a data extraction sheet. Three independent investigators collected this information and subsequently organized them in the tables. Finally, to ensure no duplications or overlap existed in the content, all the selected articles were cross-checked by other authors.

### 2.4. Quality and Risk of Bias Assessment

As aforementioned, we applied the PRISMA checklist to ensure the quality and reliability of selected articles. Two independent researchers evaluated the consistency and quality of the articles and the risk of bias. In either case of discrepancy in viewpoints, a third independent researcher resolved the issue. The full text of selected articles was read, and the key findings were extracted.

Included studies underwent quality check and risk of bias assessment. This qualitative analysis was performed according Murad’s quality checklist of case series and case report [33]. As reported, the scale consists of four parameters, to evaluate the (a) patient selection; (b) exposure ascertainment; (c) causality; (d) reporting. Each section contains one to four question to be addressed. As it is suggested we performed an overall judgement about methodological quality since questions 4, 5 and 6 are mostly relevant to cases of adverse drug events. Each requested field will be considered as adequate, inadequate or not evaluable. The table showing this tool for evaluating the methodological quality of case reports and case series, is reported in the original manuscript [33].

## 3. Results

In this study, 117 documents were identified using the systematic search strategy. After a primary review of 2201 retrieved articles, 379 duplicates were removed, and the title and abstract of the remaining 1822 resources were reviewed. After applying the selection criteria, only 117 articles met the inclusion criteria and were included in the final review (Figure 1). Therefore, the cases confirmed according to these parameters were 260 (Table 2). 

### 3.1. Demographic and Clinical Features of Reinfection Cases

Reinfection occurred across the world: 1 case from Austria, 1 from Bahrain, 5 from Bangladesh, 2 from Belgium, 31 from Brazil, 3 from China including 1 from Hong Kong, 2 from Colombia, 28 from the Czech Republic, 1 from Denmark, 2 from Ecuador, 10 from France, 2 from Gambia, 1 from Germany, 24 from India, 31 from Iran, 12 from Iraq, 1 from Israel, 5 from Italy, 1 from Japan, 1 from Lebanon, 1 from Libya, 4 from Mexico, 5 from Pakistan, 1 from Panama, 1 from Peru, 1 from Portugal, 6 from Qatar, 1 from South Korea, 1 from Switzerland, 8 from Saudi Arabia, 1 from South Africa, 9 from Spain, 1 from the Netherlands, 4 from Turkey, 9 from the United Kingdom, 42 from the United States of America (Figure 2).

Age was reported in 237 cases: 5/237 patients (2.1%) were between 0 and 20 years old, 95/237 (40%) between 21 and 40 years old, 83/237 (35%) between 41 and 60, 42/237 (17%) between 61 and 80, and 12/237 (5%) > 80 years old (Figure 3).

Gender was reported in 251/260 cases, among which 115/251 patients (45.8%) were female and 136/251 (54.2%) were male (Figure 4).

The main risk groups were healthcare workers and patients with comorbidities. In total, 66/260 cases (2.3%) occurred among high risk groups, including healthcare workers (HCWs), doctors, students and nursing resident. A total of 91 cases (35%) occurred among patients with comorbidities, 48 in men and 38 in woman (Figure 5).

The evolution of the reinfection episode itself was more severe in 92/260 (35.3%) cases with the death only in 14/260 cases (5.3%), 7/260 male (2.65%) and 7/260 females (2.65%); 8 of these had a neoplastic immune system diseases, or transplant or other important comorbidities and 3 were over 80 years old (Figure 6).

Notably, reinfection occurred among patients whose initial infections were both asymptomatic/mild, 80% (207/260), and moderate/severe, 20% (53/260). The demonstration that moderate/severe initial infections do not necessarily provide enhanced protection against reinfection is important because patients with more severe infection have been found to have higher neutralizing antibody titers, which may be expected to confer protection. Additionally of note, the severity of the reinfection episode itself was less in 21/53 cases (40%). The observation that many reinfection cases were less severe than initial cases is interesting because it may suggest partial protection from disease [152] and argues against antibody-dependent immune enhancement, which can be seen with other viral pathogens. In the absence of routine surveillance, we would have expected a bias toward detection of symptomatic reinfection, underscoring the importance of prospective screening.

Another interesting datapoint is the detection of different clades or lineages detected by genome sequencing between initial infection and reinfection in 52/260 cases (20%). The current gold standard for identifying reinfection is detection of a distinct virus by genome sequencing. Detection of reinfection is most straightforward when viruses belong to a different clade or lineage, as this provides clear evidence of infection by a different virus [6]. Although reinfection is most apparent when viruses are different enough to distinguish by genome sequencing, it remains unclear whether these viral genomic differences play a causative role in reinfection. That is, does reinfection occur when viral genomic differences permit escape from an existing, but narrow, immune response to the initial infection? Answering this question will require detailed mapping of the relationship between virus substitutions and immune escape.

### 3.2. Quality and Risk of Bias Assessment

Briefly, only 14 studies fulfilled the quality checklist. “Selection—Does the patient(s) represent(s) the whole experience of the investigator (center) or is the selection method unclear to the extent that other patients with similar presentation may not have been reported?” checklist resulted unclear in most of the studies, because the patient selection method was unclear. In general, overall quality was satisfactory in all included studies.

## 4. Discussion

Since the first cases, a question has haunted all researchers: can a patient recovered from COVID-19 get sick again? The first confirmed case of reinfection occurred in a 33-year-old Caucasian man of Hong Kong, that was admitted to the hospital for COVID-19 on 23 March 2020 [5]. After two negative tests by RT-PCR on days 21 and 22 he was discharged from the hospital and resumed his usual work [5]. Serological controls after the first infection showed that he did not produce virus neutralizing antibodies [139]. On 15 August 2020 after a 1-week trip in Spain, the patient returned to Hong Kong and was submitted to a collection of a deep throat saliva sample for RT-PCR as border surveillance and resulted positive [5]. The patient was asymptomatic until the new negative test. The viruses from the first and the second infection were phylogenetically distinct and the virus of first infection had a truncation in the 58AA open reading frame 8 gene, that could be responsible immune evasion [138]. However T cells and mucosal immunity might have played an important role in resolving the second infection, even if there was the absence of primary neutralizing antibodies [139].

In October 2020, Tillett et al. reported the first confirmed case of SARS-CoV-2 reinfection in the USA [27]. A 25-year-old man from Nevada, without known immune disorders, had PCR-confirmed SARS-CoV-2 infection in April, 2020 (cycle threshold (Ct) value 35·24; specimen A) [27]. He recovered in quarantine, testing negative by RT-PCR at two consecutive timepoints thereafter [27]. However, 48 days after the initial test, the patient tested positive again by RT-PCR (Ct value 35·31; specimen B) [27]. Viral genome sequencing showed that both specimens A and B belonged to clade 20C, a predominant clade seen in northern Nevada [27]. The genome sequences of isolates from the first infection (specimen A) and reinfection (specimen B) differed significantly, making the chance of the virus being from the same infection very small [27]. The particularity of this report is that SARS-CoV-2 reinfection resulted in worse disease than the first infection, requiring oxygen support and hospitalization [27]. The patient had positive antibodies after the reinfection, but whether he had pre-existing antibody after the first infection is unknown [27]. Both cases reported from Nevada and Hong Kong seem to confirm the possibility that the reinfections are due to a different variant of SARS-CoV-2. 

The first important question to be answered is: are all cases reported in the literature as reinfection by SARS-COV-2 true reinfections? 

A distinction must be made between true reinfection, relapsed infection, recurrence of positive (re-positive) nucleic acid detection [17,153], in fact one of the features of SARS-CoV-2 infection is prolonged virus shedding. Several studies reported persistent or recurrent elimination of viral RNA in nasopharyngeal samples starting from first contact with a positive subject [18,19,20]. Several explanations can exist in order to explain this phenomenon without it being a true reinfection. One possible explanation for testing positive after a previously negative result could be that the negative results after patient recovery were really false-negative results [154]. Literature reported that false-negative rates can be as high as 30% for SARS-CoV-2 PCR testing [155]. However, actually the KCDC (Korean Control Disease Center) determined recovery as two separate negative PCR results within 24 h [156]. In this way, patients positive after having two consecutive negative results would be positive for an increase in viral genetic material due to reinfection [156]. It is difficult to have two previous consecutive false-negative results [156]. Another possible explanation could be the contamination of the samples, but most testing centers are requiring testers to change personal protective equipment (e.g., gloves, gowns and masks) [156]. However, surely one of the main points to consider is the basis of PCR testing: the test is able to amplify nucleic acid in the sample, not fully active viral particles. The genetic material (RNA and DNA) left behind degrades over time [157]. Thus, positive PCR results after recovery may not necessarily signify reinfection, but rather the presence of leftover genetic material from previously active infection [156]. Therefore, a patient who retests positive for virus might not necessarily be experiencing a second, new SARS-CoV-2 infection [158]. True reinfection has criteria that must be considered, including isolation of the complete genome of the virus (and not just genomic fragments) in the second episode, identification of two different virus strains in two episodes of infection based on phylogenetic analysis; proof of virus infectivity in the second episode by virus isolation and evaluation of its cytopathic effect in cell culture; investigation of immune responses and their comparison in two episodes; epidemiologic data such as re-exposure history to COVID-19 patient in the second event and timing between episodes, with a longer time interval between two episodes favoring the reinfection hypothesis [17,159]. To date, positive retesting more than 83 days after the first positive test, along with other criteria, favors confirmation of reinfection, even if Turner et al. recently reported a patient with prolonged viral RNA shedding lasting 87 days after the initial positive clinical PCR test and 97 days after the onset of symptoms, probably due to the poor CD8+ T cell response during the first three months of his illness [160]. In addition to the abovementioned reasons, the disease clinical data are also useful in confirming the second episode, although the second episode may be asymptomatic [17]. A time interval where the patient is free of clinical signs between the two episodes is also necessary. In conclusion, only cases with clinical symptoms and RT-PCR positivity after negative tests following recovery from COVID-19 could be considered true SARS-CoV-2 reinfections. Recently Raveendran et al. suggested an interesting approach in order to individuate the reasons for a persistent RT-PCR positivity (Figure 7) [161]. According to this flow chart it is possible to individuate cases of persistent RT-PCR positivity due to reinfection or to presence of dead viral fragment or to persistent viral shedding.

The second important question to be answered is: can SARS-CoV-2 re-infect a patient after recovery?

When any unwanted virus comes into contact with our body, also in the case of SARS-CoV-2 infection, most patients are able to develop specific antibodies neutralizing the spike proteins of this virus [5]. A recent study of Pilz et al. pointed out that the relatively low tentative reinfection rate (40 cases in 14,840 COVID-19 survivors of first wave—0.27%) ensures a good protection after natural infection for SARS-CoV-2 [162]. However there are three main mechanisms for reinfection: the immune response can be ineffective, strain-specific, or short-lived [156].

Monoclonal antibodies formed against the SARS-CoV-2 virus target the Spike (S) glycoprotein component, the receptor-binding domain of the virion [156]. SARS-CoV-2, however, has been shown to develop “escape mutants,” or alterations, in the epitope of the S protein that contribute to host tropism and viral virulence [156]. Sui et al. reported that major variations exist in the S protein at positions 360, 479, and 487 [163]. They found that altering 1–2 amino acids at those positions led previously efficacious neutralizing antibodies to SARS-CoV-2 to a 20–50% reduction in binding capacity [163]. Theoretically, if SARS-CoV-2 is also able to form “escape mutants” in the S protein, IgG antibodies formed in patients may be less ineffective, though not completely, in neutralizing the virus [156]. This could mean that patients remain resistant to SARS-CoV-2 infection even after mutations, with antibody responses that are 50–80% efficacious [156].

Another possibility that could allow the reinfection of a patient is the duration of the body immune response [156]. Recent findings suggested that protective immunity does not occur in all infected individuals [164], supporting the possibility of reinfection [103], even if 93% of the infected produce neutralizing antibodies [165]. Their function is to prevent the virus from entering cells between 6 and 20 days after infection [166] with this mechanism: after the infection, B lymphocytes are activated and produce IgM, IgG and IgA antibodies. A subset of them (IgG and IgA) then manage to make the new viral particles harmless. The neutralizing antibodies, in turn, are accompanied by the activation of killer cells (T lymphocytes), specialized in recognizing and destroying the virus [167].

Seroconversion of IgM and IgG antibodies occurs the first week after onset of symptoms, seroconversion rates rise until the fourth week and decline thereafter, by the seventh week IgM antibodies are not detected in most cases, even if some reports showed IgM antibodies to persist for up to 8 months post-COVID-19 [168], whereas IgG antibodies persist longer for a period of time yet unknown [169]. Immunoglobulins alone are not truly sufficient to confer long-term immunity to coronavirus [156]. CD4+ T-cells and memory CD8+ T-cells with their products, such as effector cytokines and IFN-γ, are important in providing protection from coronavirus [170]. In fact, when the infection is over, in the following weeks or months, the antibodies drop: the virus is no longer there, they are no longer needed. However, the memory cells remain in the body, ready to intervene in case of need. All the studies so far show that a long-lasting immune response occurs. A very recent study carried out in collaboration between the Policlinico San Matteo in Pavia and the Karolinska Institute in Stockholm quantifies this “time” more precisely: memory cells persist for at least 6–8 months after infection [171]. Considering that the disease erupted just under a year ago, this is the maximum observation time possible to date, but it could be much longer [171]. Previous studies showed that virus-specific memory CD8+ T-cells were found to persist for up to 6 years after a SARS associated coronavirus infection, but memory B-cells and accompanying antibodies were undetectable at that time [172]. However Vetter et al. hypothesized that reinfection can be due to a loss of protection elicited after the first episode for a progressive reduction of protective antibody titers [144,173].

We can conclude that antibody formation and longevity of immunity in a subject could be dependent by the strain of virus, its severity and age of subject [174].

Khoshkam et al. tried to classify the recovered and immunized subjects in four categories:(1)Infected cases with very mild symptoms or asymptomatic without any humoral immune response or elicited memory.(2)Infected cases with mild to moderate symptoms with low humoral immunity and low cellular immunity.(3)Infected cases with moderate or severe symptoms with highly activated humoral immunity and elicited memory.(4)Infected cases with moderate or severe symptoms with highly activated humoral immunity and low cellular immunity [175].

They hypothesized that reinfection may happen in groups 1 and 2, which may also develop the severe disease in the future due to the absence or low levels of acquired immunity [175]. Individuals in group 3 are more protective against further exposures and they may show long-term immunity since they develop increased elicited memory in defense of SARS-CoV-2 [175]. The last group may show rapid response against reinfection; they may not be safe for longer periods because of the non-imprinted memory of immunity [175].

The question to be solved is whether these antibodies can neutralize each SARS-CoV-2 clade and guarantee immunity to subsequent contact. Reinfection from SARS-CoV-2 with a genetically distinct strain of SARS-CoV-2 is, in theory, possible in patients immediately after recovery from COVID-19. SARS-CoV-2 infection may not confer immunity against a different SARS-CoV-2 strain, so more research is needed. SARS-CoV-2, even if it is a virus similar to that of the flu, seems to have a more stable genome and the response that the immune system generates is towards several fragments of the viral proteins and not just one. In fact, the mutations observed so far (and, perhaps, also the new English variant, at least until proven otherwise) are not associated with a change in the severity of the disease. 

The new variants are accumulating mutations in different spike domains, such as the alpha variant or B.1.1.7 lineage (also known as 501Y.V1 or VOC202012/01), the beta variant or B.1.351 lineage (501Y.V2), the gamma variant or P.1 lineage (501Y.V3) and the delta variant or B.1.617.2 lineage [176]. All these variants have cumulated at least nine non-synonymous mutations/deletions throughout the Spike coding region. For example, the case reported by Harrington et al. showed that anti-SARS-CoV-2 antibodies were still present shortly before onset of reinfection, with no evidence of antibody waning [82]. This may raise some concerns about immune evasion by the alpha variant, which is a concern with the high number of spike region mutations seen. However, the study has a bias: there were no assays for SARS-CoV-2 antibodies recognizing spike antigen in the second reinfection, while the tested antibodies recognized “N” antigen, so it is difficult to point out an evident role of antibodies in the reinfection. The 501Y.V2 variant, or beta variant, is characterized by eight mutations in the spike protein-coding sequences that can improve its ability to transmission [151]. The case reported by Zucman et al. showed that beta variant can be more aggressive than non-VOC SARS-CoV-2 [151]. The last, the delta variant, is characterized by P681R and L452R mutations that can help the delta variant spread. For all these reasons it is necessary to investigate urgently the possibility of these new variants to escape the vaccine action. The immune responses generated by mRNA and adenoviral vector-based vaccines are restricted to the Spike glycoprotein, so new variants with big antigenic drift could reduce their efficiency and determine a growing number of reinfections.

Another possibility that could allow the reinfection of a patient is the reactivation of dormant virus which is commonly seen in immunosuppressed patients with some viruses, such as Epstein Barr, cytomegalovirus and herpes groups [90], but it is necessary to sequence viral genome for differential diagnosis between viral reactivation or reinfection with a different strain. 

For all these reasons, it is important to identify cases of reinfection to understand if the “immunological memory” affects the symptoms during a second infection, a crucial fact, in particular, to predict the effectiveness of the vaccination campaign. If in the second time the symptoms are generally reduced, as in the Hong Kong [5], Belgium and the Netherlands [29] patients, this suggests that the immune system is responding as it should. However, if symptoms are consistently more severe during a second COVID-19 attack, as in the case of the Nevada [27] or Ecuador [28] subjects, it may be that the immune system makes matters worse. The mechanisms that could account for a more severe secondary infection can only be speculated. First, a very high dose of virus might have led to the second instance of infection and induced more severe disease [177]. Second, it is possible that reinfection was caused by a more virulent variant of the virus, or more virulent in this patient’s context [27]. Third, a mechanism of antibody-dependent enhancement might be the cause, a means by which specific Fc-bearing immune cells become infected with virus by binding to specific antibodies [27]. In fact, the clinical course of some severe COVID-19 cases has been worsened by abnormal immune responses that damage healthy tissue. Patients who experienced that problem during a first infection may have immune cells that are induced to respond disproportionately the second time too. Sometimes antibodies produced in response to SARS-CoV-2 can facilitate the virus during a second infection rather than fight it [178,179,180,181,182,183,184]. The phenomenon [185,186,187,188,189] is rare, but researchers have found worrying signs of it while trying to develop vaccines against the coronaviruses responsible for severe acute respiratory syndrome and Middle East respiratory syndrome [190] and against SARS-CoV-2 [191,192,193,194].

As researchers accumulate more examples of reinfection, the situation should become clearer. Depending on the criteria used, rates of reinfection can vary widely [195]. There are some reports about retrospective observational study such as that of Pilz et al. that reported 40 cases of tentative reinfection in Austria, but these data are limited by the lack of detailed clinical characteristics [162]. For this reason, in November 2020 the Centers for Disease Control and Prevention pointed out the following criteria to define reinfection with SARS-CoV-2: detection of SARS-CoV-2 RNA (with Ct values < 33 if detected by RT-PCR) >90 days after the first detection of viral RNA whether or not symptoms were present and paired respiratory specimens from each episode that belong to different clades of virus or have genomes with >2 nucleotide differences per month [32]. Cases in which detection of SARS-CoV-2 RNA is present >45 days to 89 days apart are considered reinfections if the second symptomatic episode had no obvious alternate explanation for the COVID-19-like symptoms or there was close contact with a person known to have laboratory diagnosed COVID-19 and paired specimens are available with the Ct values and sequence diversity noted above. 

However, the ability to re-infect does not mean that a SARS-CoV-2 vaccine cannot be effective. Some vaccines, for example, require a “booster” dose to maintain protection. Learning more about reinfection could help researchers in developing truly effective vaccines by showing them which immune responses are important for maintaining immunity. For example, researchers may find that people become vulnerable to reinfection after antibodies drop below a certain level, and so they can modify vaccination strategies accordingly using a booster dose to maintain that level of antibodies. At a time when health authorities are grappling with the dizzying logistical difficulties of vaccinating the world population against SARS-CoV-2, the need for a booster injection is a necessity that complicates the management of the vaccination campaign, but it does not make long-term immunity from SARS-CoV-2 impossible. However, some researchers fear that vaccines will only reduce symptoms during a second infection, rather than prevent it altogether. While giving some advantages, this possibility could turn vaccinated individuals into asymptomatic carriers of SARS-CoV-2, putting vulnerable populations at risk. The elderly, for example, are among the most affected by COVID-19, but they tend not to respond well to vaccines. For all these reasons, it would be interesting to see data on how much virus SARS-CoV-2 reinfected individuals spread. 

The real problem to be solved is, therefore, the duration of immunity conferred by a COVID-19 episode. There is evidence in the literature that the COVID-19 immune response is variable and patient-specific with respect to the development of antibodies and to antibody persistence in serum over time [146]. In considering the protective effect of antibodies against a reinfection, the evidence is still inadequate, and more research is necessary in order to clarify the interplay between the roles of adaptive and innate immunity. A recent study of Gudbjartsson et al. reported that Icelandic humoral response to SARS-CoV-2 infection was persistent within the 120-day timeframe used with a modest decline in antibody titers after 120 days [196]. Iyer et al. observed declining antibody titers over 90 days, with *“median times to sero-reversion of 71 and 49 days following symptom onset”* [197].

The genetic analysis of all the new cases reported as reinfection would help in understanding if the reinfection would be due to a new infection by a different SARS-CoV-2 or a reinfection by the same virus for a decline of immune response, but unfortunately genomic analysis is not available for some of these cases. 

## 5. Conclusions

All these findings are useful and contribute towards the role of vaccination in response to the COVID-19 infections. Collected data show a wide range of situations: spanning a broad distribution of ages, risk groups, baseline health status and reinfection severity compared to the initial infection. Reinfection occurred as early as 45 days or >300 days after the initial infection. Common explanations for reinfection can be either waning SARS-CoV-2 antibodies or the presence of viral escape mutations [198]. While several cases of SARS-CoV-2 reinfection did involve infection with a different clade, it is noteworthy that mutations were identified throughout the genomes and the frequency of mutations within the S gene was not elevated relative to the rest of the genome [199]. In addition, individuals with more severe reinfections did not have significantly greater frequency of S gene mutations [199]. Finally, the presence of rare mutations was uncommon in the re-infecting virus, which largely mirrored the contemporaneously circulating variants in the region of infection, as reported by Choudhary et al. [199]. Concerning the problem of recognizing reinfection and persistent infection, two factors generally differentiated them. First, reinfections have so far been largely described in immunocompetent individuals while the majority of persistent COVID cases have been in immunosuppressed patients [199]. Secondly, phylogenetic analysis can generally differentiate between reinfection and persistent infection, especially in cases where persistent infection allowed the longitudinal collection of >2 sequences [199]. Due to the reinfection cases with SARS-CoV-2, it is evident that the level of immunity is not 100% for all individuals. Reinfection with SARS-CoV-2 is a possibility in both vaccinated and unvaccinated individuals, because vaccines to the virus may not translate to total immunity [199]. Recently breakthrough infections were reported following mRNA vaccination in healthy subjects [200,201], despite evidence of effective immune response among the breakthrough subjects [202]. Another study reported that eight symptomatic SARS-CoV-2 infections occurred in fully vaccinated healthcare workers (incidence rate 4.7 per 100,000 person-days adjusted) [203]. This type of challenge was also observed during the process of vaccine preparation for influenza [204]. Even though several vaccines are ready, the presence of more than 80 genotypical variants of the virus, possibility of reinfection, and short duration of seropositivity for neutralizing antibodies raise the concern that vaccination may not result in an effective and long-term immunity against SARS-CoV-2. Furthermore, immunoglobulin levels may not correlate with viral shedding and risk of transmissibility of SARS-CoV-2 [205]. Additionally, the short duration of immunity against the virus may not allow for increasing homogeneity of affected populations in a non-specific time frame. These factors raise concerns that eliminating the COVID-19 pandemic may not be as feasible as once assumed and that we must rely more on prevention of transmission until more aspects of the virus and its pathogenicity are discovered. A recent study suggested that among persons with previous SARS-CoV-2 infection, full vaccination provides additional protection against reinfection [206]. In fact, among previously infected Kentucky residents, those who were not vaccinated were more than twice as likely to be reinfected compared with those with full vaccination [206]. Data from literature are comforting: out of hundreds of millions of people infected with the virus and then cured, only a few are reported cases of confirmed reinfection [199]. Despite the appearance of different variants of the virus, vaccines seem to help us for the near future. However, the presence of immunosuppressed or transplanted subjects requires us to continue to observe the precautionary rules useful to prevent the spread of the virus. In fact, it is imperative that all individuals, whether previously diagnosed with COVID-19 or not should take identical precautions to avoid reinfection with SARS-CoV-2 till the time when community immunity had been achieved [207]. All eligible persons should be offered vaccination, including those with previous SARS-CoV-2 infection, to reduce their risk for future infection [206].

This report highlights how it is necessary to continue to observe all the prescriptions recently indicated in the literature [208,209,210] in order to avoid new contagion for all patients after healed from COVID-19 or asymptomatic positive, since the infection does not ensure complete immunity in 100% of cases.

## Figures and Tables

**Figure 1 ijerph-18-11001-f001:**
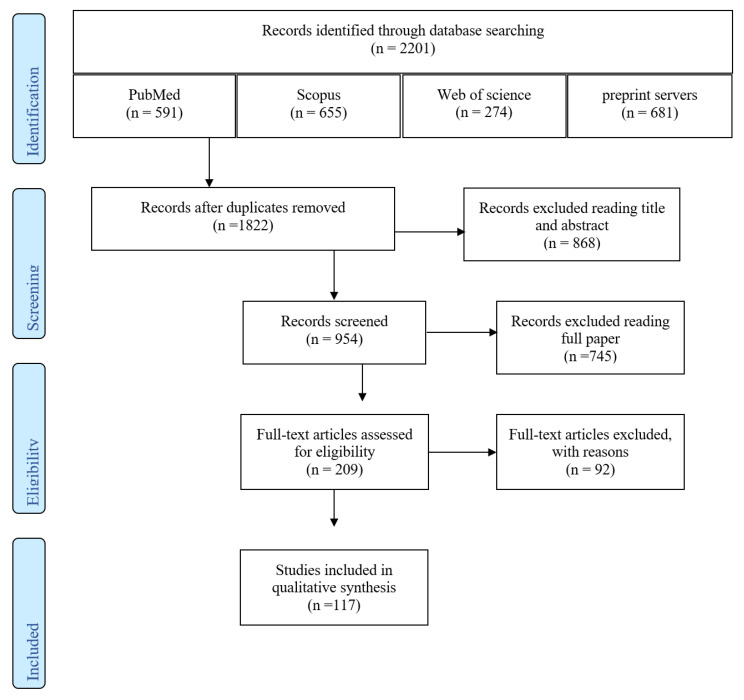
Flow diagram for the selection process of identified articles.

**Figure 2 ijerph-18-11001-f002:**
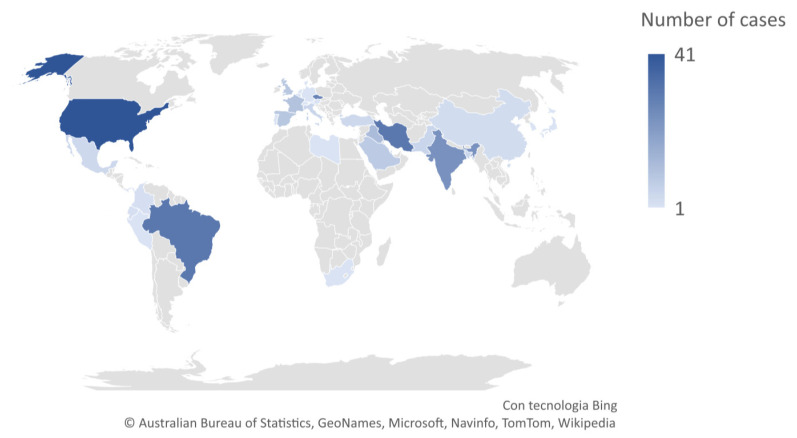
Distribution of cases worldwide.

**Figure 3 ijerph-18-11001-f003:**
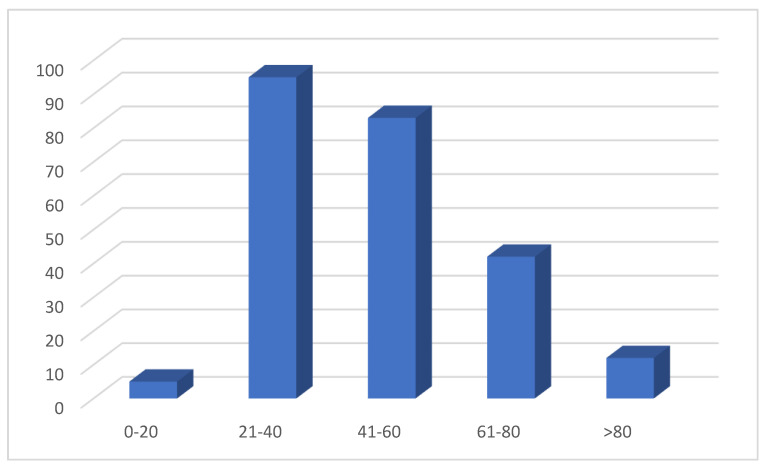
Distribution of cases according age.

**Figure 4 ijerph-18-11001-f004:**
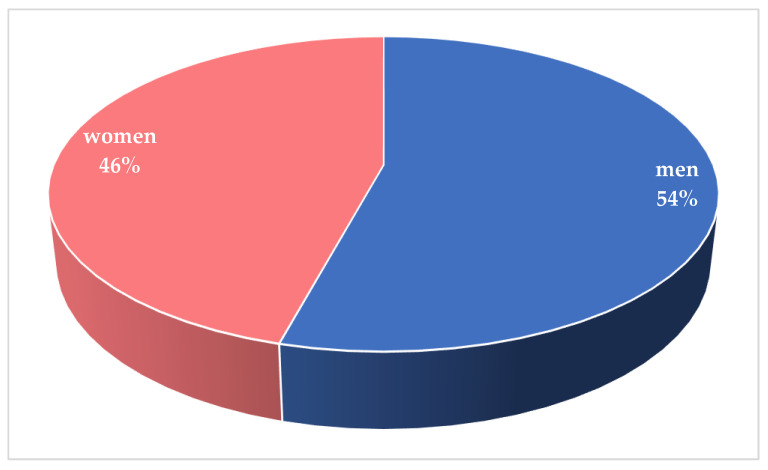
Distribution of cases according sex.

**Figure 5 ijerph-18-11001-f005:**
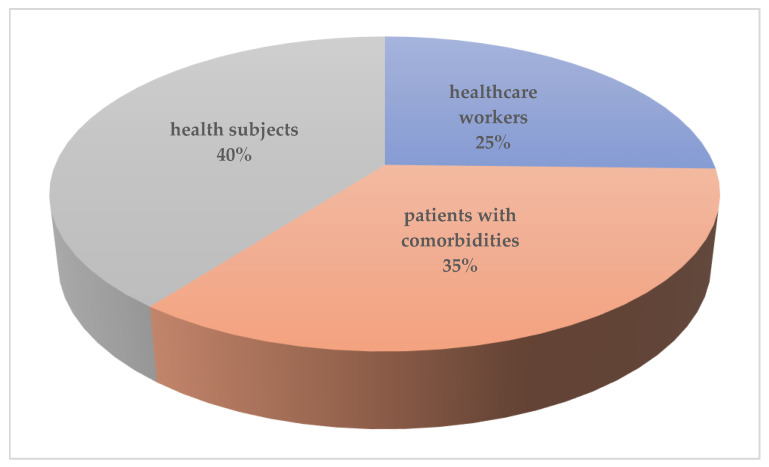
In total, 60% of reinfection involved patients in risk groups.

**Figure 6 ijerph-18-11001-f006:**
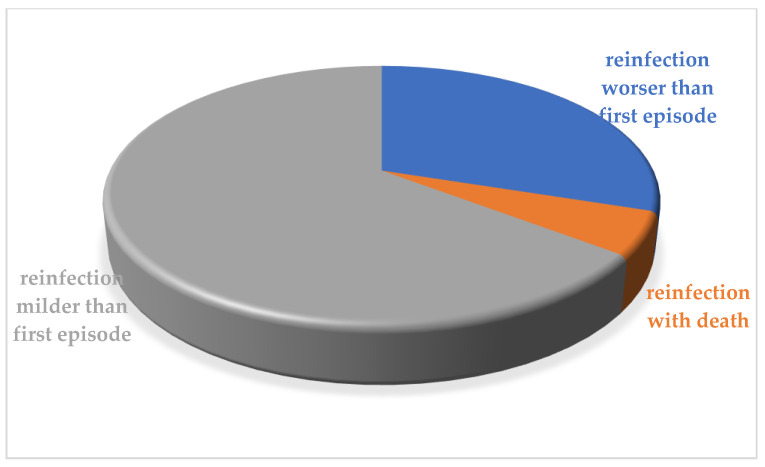
The evolution of the reinfection episode was more severe in 35.3% of cases.

**Figure 7 ijerph-18-11001-f007:**
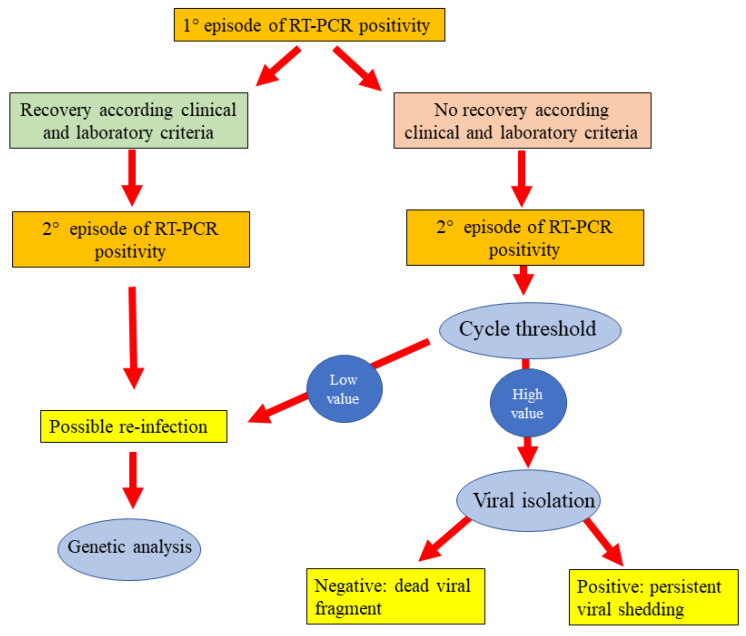
Flow diagram in order to determine the cause of persistent RT-PCR positivity for SARS-CoV-2, modified by Raveendran, A.V. et al. [161].

**Table 1 ijerph-18-11001-t001:** Protocol of Center for Disease Control and Prevention for investigating suspected SARS-CoV-2 reinfection.

Investigative Criteria	Laboratory Evidence
People with detected SARS-CoV-2 RNA (if detected by RT-PCR, only include if Ct value < 33 or if Ct value unavailable) ≥90 days after the first detection of SARS-CoV-2 RNA, whether or not symptoms were present	*Best evidence* Differing clades as defined in Nextstrain and GISAID of SARS-CoV-2 between the first and second infection, ideally coupled with other evidence of actual infection (e.g., high viral titers in each sample or positive for subgenomic mRNA, and culture)
2. People with detection of SARS-CoV-2 RNA (if detected by RT-PCR, only include if Ct value < 33 or if Ct value unavailable) ≥45 days after the first detection of SARS-CoV-2 RNA **AND** with a symptomatic second episode and no obvious alternate etiology for COVID-19-like symptoms or close contact with a person known to have laboratory-confirmed COVID-19	*Moderate evidence* >2 nucleotide differences per month ^*^ in consensus between sequences that meet quality metrics above, ideally coupled with other evidence of actual infection (e.g., high viral titers in each sample or positive for subgenomic mRNA, and culture)
	*Poor evidence but possible* ≤2 nucleotide differences per month * in consensus between sequences that meet quality metrics above or >2 nucleotide differences per month ^*^ in consensus between sequences that do not meet quality metrics above, ideally coupled with other evidence of actual infection (e.g., high viral titers in each sample or positive for subgenomic mRNA, and culture)

* The mutation rate of SARS-CoV-2 is estimated at 2 nucleotide differences per month, therefore if suspected reinfection occurs 90 days after initial infection, moderate evidence would require >6 nucleotide differences.

**Table 2 ijerph-18-11001-t002:** Cases of SARS-CoV2 reinfection in the international literature (all cases were again positive for SARS-CoV-2 after complete symptomatic recovery in addition to negative RT-PCR test for SARS-CoV-2, according to WHO recommendations [34]).

Authors	Year	Patient Country	Patient	Interval Time between 1 Infection and Reinfection	Viral Genome Sequence	COVID-19	Symptoms	Antibody after First Infection or Reinfection
Abu-Raddad LJ et al. [35]—case 27	2021	Qatar	25–29-year-old man	46	9 SNVs compared to initial infection strain, including D614G	Mild	N/A	N/A
						Mild	N/A	
2. Abu-Raddad LJ et al. [35]—case 33	2021	Qatar	40–44-year-old man	71	11 SNVs compared to initial infection strain, including D614G	Mild	N/A	N/A
						Mild	N/A	
3. Abu-Raddad LJ et al. [35]—case 20	2021	Qatar	45–49-year-old woman	88	3 SNVs compared to initial infection strain, including D614G	Mild	N/A	ROCHE elecsys antiSARS-CoV-2 negative at time of reinfection
						Mild	N/A	
4. Abu-Raddad LJ et al. [35]—case 44	2021	Qatar	25–29-year-old woman	55	1 SNVs compared to initial infection strain, including D614G	Mild	N/A	N/A
						Mild	N/A	
5. Adrielle dos Santos L et al. [36]	2021	Brazil	44-year-old healthcare man with systemic arterial hypertension, obesity	53	20A	Mild	Dry cough, dyspnea, dysgeusia, diarrhea, asthenia, sneezing/runny nose	N/A
					Clade B.1.1.28	Worse	Dry cough, dyspnea, fever, myalgia, asthenia, arthralgia, headache, nausea/vomiting, sneezing/runny nose, severe respiratory symptoms and was admitted to ICU, dying after 20 days of symptoms	
6. Aguilar-Shea AL et al. [37]	2021	Spain	39-year-old healthcare man	290	N/A	Mild	Sore throat, fever, general malaise, nasal congestion, tachycardia, chest pain, loss of smell and taste	Rapid antibody test: positive
					201/501Y.V1.Britain variant B.1.17	Milder	Sore throat, slight general malaise, nasal congestion, tiredness	Rapid antibody test: positive
7. Ahmadian S et al. [38]	2021	Iran	36-year-old healthcare man	60	N/A	Mild	Lethargy, fatigue, shortness of breath, headache, fever, chills	N/A
						Milder	Eye infection, fever, fatigue, shortness of breath, muscle pain	
8. Ahmed A et al. [39]	2021	Pakistan	Healthcare worker man	118	N/A	Mild	Arthralgia, weakness, anosmia, ageusia	N/A
						Milder	Fever, sore throat, dry cough	
9. Ahmed A et al. [39]	2021	Pakistan	Healthcare worker man	86	N/A	Mild	Fever, sore throat	N/A
						Milder	Sinusitis	
10. Ak R et al. [40]	2021	Pakistan	40-year-old male	94	N/A	Mild	Fever	N/A
						Worse	Sore throat, cough, diarrhea	
11. Aldossary B et al. [41]	2021	Bahrain	47-year-old woman without comorbidities	60	N/A	Mild	Mild respiratory tract symptoms	N/A
						Worse	Abdominal pain, fulminant hepatic failure > death	
12. Ali A. et al. [42] Patient 1	2020	Iran	20s year age range, male	89 **	N/A	Mild	Fever, myalgia	6.7 IgG (s/ca) after recovery
						Worse	Fever, myalgia, cough, loss of taste, loss of smell	
13. Ali A. et al. [42] Patient 2	2020	Iran	30s year age range, female	55 **	N/A	Mild	Fever, myalgia	10.3 IgG (s/ca) after recovery
						Worse	Fever, loss of taste and smell, myalgia, cough	
14. Ali A. et al. [42] Patient 5	2020	Iran	40s year age range, male	55 **	N/A	Mild	Fever, myalgia	15.5 IgG (s/ca) after recovery
						Mild	Fever, myalgia, cough	
15. Ali A. et al. [42] Patient 8	2020	Iran	50s year age range, male	46 **	N/A	Mild	Fever, myalgia	10.3 IgG (s/ca) after recovery
						Worse	Fever, loss of taste and smell, myalgia, cough	
16. Ali A. et al. [42] Patient 9	2020	Iran	50s year age range, female	53 **	N/A	Mild	Fever, loss of taste and smell	5.35 IgG (s/ca) after recovery
						Milder	Fever, myalgia, cough	
17. Ali A. et al. [42] Patient 11	2020	Iran	40s year age range, male	76 **	N/A	Mild	Fever, myalgia	7.22 IgG (s/ca) after recovery
						Worse	Fever, loss of taste and smell, myalgia, cough	
18. Ali A. et al. [42] Patient 12	2020	Iran	40s year age range, female	45 **	N/A	Mild	Fever, myalgia	11.2 IgG (s/ca) after recovery
						Worse	Fever, loss of taste and smell, myalgia, cough	
19. Ali A. et al. [42] Patient 14	2020	Iran	40s year age range, male	50 **	N/A	Mild	Fever, loss of taste and smell, myalgia	12.51 IgG (s/ca) after recovery
						Mild	Fever, loss of taste and smell, myalgia, cough	
20. Ali A. et al. [42] Patient 16	2020	Iran	40s year age range, male	62 **	N/A	Mild	Fever, cough	7.11 IgG (s/ca) after recovery
						Worse	Fever, loss of taste and smell, myalgia, cough	
21. Ali A. et al. [42] Patient 17	2020	Iran	40s year age range, female	49 **	N/A	Mild	Fever	8.37 IgG (s/ca) after recovery
						Worse	Fever, loss of taste and smell, myalgia	
22. Ali A. et al. [42] Patient 18	2020	Iran	40s year age range, male	72 **	N/A	Mild	Fever	5.11 IgG (s/ca) after recovery
						Worse	Fever, loss of taste and smell, myalgia, cough	
23. Ali A. et al. [42] Patient 20	2020	Iran	30s year age range, male	59 **	N/A	Mild	Fever, loss of taste and smell, myalgia	6.3 IgG (s/ca) after recovery
						Mild	Fever, loss of taste and smell, myalgia, cough	
24. Ali A. et al. [42] Patient 22	2020	Iran	50s year age range, male	53 **	N/A	Mild	Fever, myalgia	9.3 IgG (s/ca) after recovery
						Worse	Fever, loss of taste and smell, myalgia, cough	
25. Ali A. et al. [42] Patient 23	2020	Iran	20s year age range, male	49 **	N/A	Mild	Fever, myalgia	7.25 IgG (s/ca) after recovery
						Worse	Fever, loss of taste and smell, myalgia, cough	
26. Ali A. et al. [42] Patient 24	2020	Iran	40s year age range, female	52 **	N/A	Mild	Fever, myalgia	6.21 IgG (s/ca) after recovery
						Worse	Loss of taste and smell, myalgia	
27. Ali A. et al. [42] Patient 25	2020	Iran	20s year age range, female	54 **	N/A	Mild	Fever	11.9 IgG (s/ca) after recovery
						Mild	Fever, cough	
28. Ali A. et al. [42] Patient 26	2020	Iran	30s year age range, male	138 **	N/A	Moderate	Fever, loss of taste and smell, myalgia, cough	2.08 IgG (s/ca) after recovery
						Asymptomatic	Asymptomatic	
29. AlFehaidi A et al. [43]	2020	Qatar	46-year-old woman with mild asthma	80	N/A	Mild	Sore throat	N/A
						Moderate	Chest pain, fever, sore throat, body pain, cough, mild dyspnea	
30. Alshukairi AN et al. [44]	2021	Saudi Arabia	51-year-old woman with autologous hematopoietic stem cell transplantation for follicular non-Hodgkin lymphoma	160	19B	Mild	Fever, cough, malaise, and headache	Negative COVID-19 serology after 1st infection and reinfection
					20B	Mild	Fever and dyspnea	
31. Amikishiyes S et al. [16]	2021	Turkey	34-year-old man with chronic glomerulonephritis	>150	N/A	Mild	Asymptomatic	N/A
						Worse	Cough, fever, bilateral infiltrates at computed chest tomography	
32. Amorin MR et al. [45]	2021	Brazil	35-year-old healthcare worker woman	55	N/A	Mild	Fever, headache, chills, sneezing, coryza, myalgia	N/A
						Mild	Headache, nasal congestion, odynophagia, ageusia, anosmia	
33. Amorin MR et al. [45]	2021	Brazil	61-year-old healthcare worker woman with chronic bronchitis	170	N/A	Mild	Headache, cough, myalgia, odynophagia, coryza, diarrhea, ageusia	N/A
						Mild	Cough, myalgia, odynophagia, anosmia, diarrhea	
34. Amorin MR et al. [45]	2021	Brazil	40-year-old healthcare worker woman	131	N/A	Mild	Nasal congestion, coryza, cough, ageusia	N/A
						Mild	Odynophagia, sneezing, coryza, diarrhea, ageusia, anosmia	
35. Amorin MR et al. [45]	2021	Brazil	40-year-old healthcare worker woman	148	N/A	Mild	Fever, headache, myalgia, coryza, dry cough, vomiting, malaise	N/A
						Mild	Odynophagia, dry cough, myalgia, malaise, coryza, headache	
36. Arteaga-Livias K et al. [46].	2021	Peru	42-year-old healthcare worker woman	107	N/A	Mild with home management	Odynophagia, headache, malaise, rhinorrhea, ageusia, anosmia, cough	IgM and IgG+
						Worse with home management	Chest pain, productive cough, anosmia, pneumonia	
37. Atici S et al. [47]	2021	Turkey	46-year-old healthcare worker man	114	N/A	Moderate	Fever, sore throat, headache, cough, weakness, nausea and diarrhea, bilateral ground glass opacities and peribronchial thickening predominating on the right lung	N/A
						Mild	Sore throat, fever, headache, myalgia, weakness and nausea	
38. Atici S et al. [47]	2021	Turkey	47-year-old healthcare worker woman	128	N/A	Mild	Myalgia, headache and abdominal pain started without fever and cough	N/A
						Worse	Sore throat, headache and myalgia, fever, cough and mild respiratory symptoms, ground glass opacities and subpleural nodule on the left lung base consistent with COVID-19 on chest CT imagine	
39. Awada H et al. [48]	2021	Lebanon	27-year-old man	56	N/A	Mild	Fever, chills, diffuse arthralgia, myalgia, headache, back pain	N/A
						Milder	Fever, headache	
40. Bader N et al. [49]	2021	USA	73-year-old man with obesity, chronic obstructive pulmonary disease, pancreatic insufficiency, type II diabetes mellitus	60	N/A	Mild	Shortness of breath	N/A
						Worse	Dyspnea, fevers, confusion with worsening clinical situation and intubation	
41. Baiswar S et al. [50]	2021	USA	28-year-old male with diabetes mellitus type 1, hypertension, and end-stage renal disease on hemodialysis with multiple past admissions for diabetic ketoacidosis and uncontrolled hypertension	122	N/A	Mild	Nausea and vomiting	N/A
						Worse	Headaches and altered mental status, left-hand weakness. The patient became unresponsive and was intubated for airway protection > cerebrovascular accident	
42. Bellesso M et al. [51]	2021	Brazil	76-year-old female with end-stage kidney disease related to lambda light chain multiple myeloma	126	N/A	Moderate	Hip pain, confusion, respiratory distress	N/A
						Worse	Dyspnea, acute respiratory failure, hypoxemia > death	
43. Bongiovanni M. [52]	2020	Italy	48-year-old nurse female	90	N/A	Mild	Dry cough, mild fever	LIASON ^®^ SARS-CoV-2 S1/S2 IgG+ 30 Au/mL
						Asymptomatic	Asymptomatic	IgG+ 102.9 Au/mL
44. Bonifacio LP et al. [53]	2020	Brazil	24-year-old white female without comorbidities	76	N/A	Mild with complete resolution at home within 10 days	Headache, malaise, adynamia, feverish sensation, sore throat, nasal congestion	N/A
						Worse with home resolution in 12 days, headache and hyposmia for 63 days	Malaise, myalgia, severe headache, fatigue, weakness, feverish sensation, sore throat, anosmia, dysgeusia, diarrhea, coughing	IgG/IgM– at NAAT+IgG/IgM+ 28 days after NAAT+
45. Borgogna C et al. [54]	2021	Italy	52-year-old man with transitional cell carcinoma of the renal pelvis	110	Clade 20B and Pangolin lineage B.1.1	Mild	Cough, fever	
					Clade 20A and Pangolin lineage B.1	Milder	Fever	Very low levels of IgG anti-SARS-CoV-2 Spike protein, positive IgG anti-SARS-CoV-2 N protein
46. Brehm TT et al. [55]	2021	Germany	27-year-old female nurse	282	HH-24.I (19A)	Mild	Fever, chills, dyspnea	IgG anti-SARS-CoV-2 Spike protein: 40 AU/mL in July 2020, 15 AU/mL in September 2020
					HH-24.II (20EU1) with differences in 21 positions, including 2 typical variations in spike proteins A222V and D614G	Milder	Dry cough, mild rhinorrhea	IgG anti-SARS-CoV-2 Spike protein: 97 AU/mL on 29 December
47. Buddingh EP et al. [56]	2021	The Netherlands	16-year-old girl	390	Classic	Moderate	High fever, mild conjunctivitis, malaise, chest pain, coughing, abdominal pain and diarrhea. She was diagnosed with myocarditis, shock and had high inflammatory parameters.	IgG SARS-CoV-2 was negative (Abbott SARS-CoV-2 IgG; Abbott Laboratories)
					B.1.1.7 variant (UK variant),	Mild	Mild respiratory symptoms	
48. Caralis P. [57]	2021	USA	60 with diabetes	72	N/A	Mild	Acute renal failure	
						Milder	Fatigue	
49. Caralis P. [57]	2021	USA	27 with psoriatic arthritis	79	N/A	Mild	Fever, flu-like	IgG+
						Milder	Fatigue, loss taste	
50. Caralis P. [57]	2021	USA	33 year-old woman with allergic rhinitis	172	N/A	Mild	Fever, cough, diarrhea	IgG+
						Milder	Fever headache	
51. Caralis P. [57]	2021	USA	71 with renal/liver transplant HIV, diabetes	93	N/A	Moderate	Fever, pneumonia, respiratory insufficiency	
						Asymptomatic	Asymptomatic	
52. Caralis P. [57]	2021	USA	72 with pulmonary/cardiac sarcoidosis	111	N/A	Mild	Dyspnea, fatigue, headache	
						Milder	Fatigue	
53. Cavanagaugh AM et al. [58]	2021	USA	M (80–89 years old)	101	N/A	Asymptomatic	asymptomatic	N/A
						Mild	Lethargy, decreased appetite, dry cough for 14 days	
54. Cavanagaugh AM et al. [58]	2021	USA	F (80–89 years old)	103	N/A	Asymptomatic	asymptomatic	N/A
						Worse	Congestion, respiratory failure and death	
55. Cavanagaugh AM et al. [58]	2021	USA	F (60–69 years old)	109	N/A	Mild	nausea	N/A
						Mild	Cough, sore throat, loss of appetite, malaise, muscle aches for 17 days	
56. Cavanagaugh AM et al. [58]	2021	USA	F (70–79 years old)	109	N/A	Mild	Gastrointestinal symptoms for 17 days	N/A
						Milder	Loss of appetite, malaise for 12 days	
57. Cavanagaugh AM et al. [58]	2021	USA	Female (90–99 years old)	110	N/A	Asymptomatic	asymptomatic	N/A
						Mild	Cough, loss of appetite, malaise, muscle aches for 6 days	
58. Colson P et al. [59]	2021	France	70-year-old man	105	Clade 20A	Moderate	Fever, cough	IgG+ on D26
					20A.E2, 34 nucleotide differences	Asymptomatic, during a systematic screening	Asymptomatic	
59. Das P et al. [60]—case 1	2021	Bangladesh	A 35–49-year-old man with hypertension	98	N/A	Mild	Fever, cough	
						Milder	Fever, cough, cold	
60. Das P et al. [60]—case 2	2021	Bangladesh	A 35–49-year-old researcher woman	92	N/A	Mild	Malaise	
						Milder	Sore throat, fever, cough, headache	
61. Das P et al. [60]—case 3	2021	Bangladesh	35–49 hypertensive physician	94	N/A	Mild	Fever, headache, sore throat	
						Mild	Fever, cold, low oxygen saturation	
62. Das P et al. [60]—case 4	2021	Bangladesh	35–49 man with asthma	93	N/A	Mild	Fever	
						Mild	Fever, cough	
63. Das P et al. [60]—case 5	2021	Bangladesh	35–49-year-old health worker woman with hypertension, hypothyroidism	131	N/A	Mild	Fever, cough	
						Worse	Chest pain, headache, sore throat, hospitalized	
64. Daw MA et al. [61]	2021	Libya	52-year-old healthy male	72	N/A	Mild	Cough, sore throat, fever, myalgias, headache	N/A
						Worse	Fever, cough, shortness of breath, gastrointestinal symptoms	
65. De Brito C. et al. [62]	2020	Brazil	40-year-old male doctor	46	N/A	Moderate	Fever, cough, sore throat, fatigue, myalgia, headache, diarrhea	IgG and IgM– 42 days after 1 infection
						Moderate	Fever, cough, sore throat, fatigue, myalgia, headache, diarrhea, anosmia and dysgeusia	IgG and IgM–

66. Diaz Y et al. [63]	2021	Panama	36-year-old man without comorbidities	181	A.2.4	Mild	Myalgia, chest pain, fever, cephalea, rhinorrhea, hyposmia, ageusia	
					A.2.5 containing Spike mutations D614G and L452R	Milder	Cephalea, myalgia, rhinorrhea	
67. Dimeglio C et al. [64]	2021	France	25-year-old female healthcare worker	>90	N/A	Asymptomatic	Asymptomatic	No neutralizing antibodies
						Moderate	Fever, rhinorrhea, dyspnea, chest pain, dysgeusia, anosmia, asthenia, myalgia, eye pain, pharyngitis; not hospitalized	Yes, neutralizing antibodies
68. Dimeglio C et al. [64]	2021	France	40-year-old female healthcare worker	>90	N/A	Asymptomatic	Asymptomatic	No neutralizing antibodies
						Asymptomatic	Asymptomatic	No neutralizing antibodies
69. Dimeglio C et al. [64]	2021	France	46-year-old female healthcare worker	>90	N/A	Moderate	Fever, rhinorrhea, cough, dyspnea, chest pain, intestinal disorders, dysgeusia, anosmia, asthenia, headache, myalgia, not hospitalized	Yes, neutralizing antibodies
						Mild	Fever, cough, dyspnea, chest pain, headache, asthenia, myalgia, pharyngitis; not hospitalized	Yes, neutralizing antibodies
70. Dimeglio C et al. [64]	2021	France	31-year-old male healthcare worker	>90	N/A	Mild	Anosmia; not hospitalized	Yes, neutralizing antibodies
						Asymptomatic	Asymptomatic	Yes, neutralizing antibodies
71. Dimeglio C et al. [64]	2021	France	50-year-old female healthcare worker	>90	N/A	Asymptomatic	Asymptomatic	Yes, neutralizing antibodies
						Mild	Cough, headache; not hospitalized	Yes, neutralizing antibodies
72. Dobano C et al. [65]	2021	Spain	29-year-old female healthcare worker	212	N/A	Mild	60 days	Seronegative after 1st infection, seroconverted after re-infection
						Mild	70 days	
73. Dobano C et al. [65]	2021	Spain	41-year-old female healthcare worker	154	N/A	Mild	61 days	Seronegative after 1st infection, seroconverted after re-infection
						Milder		
74. Dobano C et al. [65]	2021	Spain	58-year-old female healthcare worker	58	N/A	Mild	3 days	Unknow after 1st infection, seropositive after reinfection
						Mild	3 days	
75. Dobano C et al. [65]	2021	Spain	44-year-old female healthcare worker	211	N/A	Mild	11 days	Seropositive after 1st infection with antibody low-level
						Asymptomatic	Asymptomatic	
76. Duggan NM et al. [66]	2020	USA	82-year-old male with Parkinson, insulin-dependent diabetes, chronic kidney disease, hypertension	48	N/A	Severe with intubation	Fever, shortness of breath, hypoxia, pneumonia	N/A
						Severe without intubation	Fever, hypoxia, hypotension, tachycardia, pneumonia	
77. Elzein F et al. [67]	2021	Saudi Arabia	51-year-old man without comorbidities	58		Asymptomatic	Asymptomatic	7.04 SARS-CoV-2 IgG (Abbot) during second admission
						Worse	Fever, cough, generalized weakness, and shortness of breath, bilateral diffuse patchy airspace disease while a CT scan revealed bilateral patchy 4 central and peripheral ground glass opacities most likely related to COVID-19	
78. Elzein F et al. [67]	2021	Saudi Arabia	55-year-old man with relapsed NHL	31		Mild	Mild	0.01 SARS-CoV-2 IgG (Abbot) index negative during second admission
						Worse	High grade fever, dry cough, sore throat, tachycardia and (SPO2) 93% on room air	
79. Elzien F et al. [67]	2021	Saudi Arabia	60-year-old man with diabetes mellitus, hypertension, ischemic heart disease	27		Mild	Mild	N/A
						Milder	Cough, shortness of breath	
80. Elzein F et al. [67]	2021	Saudi Arabia	48-year-old woman with metastatic breast cancer	85		Moderate	Pneumonia	N/A
						Mild	Fever, shortness of breath	
81. Fageeh H et al. [68]	2021	Saudi Arabia	24-year-old male dental student	90	N/A	Mild	Sore throat, cough, headache, nausea, diarrhea, loss of taste and smell, insomnia, loss of appetite, and fatigue, fear and anxiety, increased insomnia, and increased body ache	N/A
						Mild	Coughing, body ache, loss of taste and smell, and diarrhea symptoms were slightly less severe, the patient was less anxious and slept well. Fever	
82. Fabianova K et al. [69]—case 1	2021	Czech Republic	60-year-old man with diabetes	177	N/A	Mild	Mild—long term care facility	N/A
						Moderate	Mild—hospitalized	
83. Fabianova K et al. [69]—case 2	2021	Czech Republic	75-year-old man with diabetes, cardiovascular disease	102	N/A	Mild	Mild—long term care facility	N/A
						Severe	Mild—hospitalized	
84. Fabianova K et al. [69]—case 3	2021	Czech Republic	72-year-old man with malignity	205	N/A	Mild	Mild—home	N/A
						Mild	Mild—home	
85. Fabianova K et al. [69]—case 4	2021	Czech Republic	62-year-old woman with asthma	137	N/A	Mild	Mild—home	N/A
						Mild	Mild—home	
86. Fabianova K et al. [69]—case 5	2021	Czech Republic	57-year-old woman without comorbidities	203	N/A	Mild	Mild—home	N/A
						Mild	Mild—home	
87. Fabianova K et al. [69]—case 6	2021	Czech Republic	56-year-old woman without comorbidities	216	N/A	Mild	Mild—home	N/A
						Mild	Mild—home	
88. Fabianova K et al. [69]—case 7	2021	Czech Republic	55-year-old man without comorbidities	212	N/A	Mild	Mild—home	N/A
						Mild	Mild—home	
89. Fabianova K et al. [69]—case 8	2021	Czech Republic	53-year-old man without comorbidities	214	N/A	Mild	Mild—home	N/A
						Mild	Mild—home	
90. Fabianova K et al. [69]—case 9	2021	Czech Republic	50-year-old woman with malignity	197	N/A	Mild	Mild—home	N/A
						Mild	Mild—home	
91. Fabianova K et al. [69]—case 10	2021	Czech Republic	49-year-old woman without comorbidities	195	N/A	Mild	Mild—home	N/A
						Mild	Mild—home	
92. Fabianova K et al. [69]—case 11	2021	Czech Republic	49-year-old woman without comorbidities	200	N/A	Mild	Mild—home	N/A
						Mild	Mild—home	
93. Fabianova K et al. [69]—case 12	2021	Czech Republic	47-year-old man without comorbidities	141	N/A	Mild	Mild—home	N/A
						Moderate	Mild—home	
94. Fabianova K et al. [69]—case 13	2021	Czech Republic	47-year-old man without comorbidities	206	N/A	Mild	Mild—home	N/A
						Mild	Mild—home	
95. Fabianova K et al. [69]—case 14	2021	Czech Republic	46-year-old man without comorbidities	154	N/A	Mild	Mild—home	N/A
						Mild	Mild—home	
96. Fabianova K et al. [69]—case 15	2021	Czech Republic	46-year-old woman without comorbidities	231	N/A	Mild	Mild—home	N/A
						Mild	Mild—home	
97. Fabianova K et al. [69]—case 16	2021	Czech Republic	45-year-old woman without comorbidities	101	N/A	Mild	Mild—home	N/A
						Mild	Mild—home	
98. Fabianova K et al. [69]—case 17	2021	Czech Republic	45-year-old woman with diabetes, chronic pulmonary disease, allergy	196	N/A	Mild	Mild—home	N/A
						Mild	Mild—home	
99. Fabianova K et al. [69]—case 18	2021	Czech Republic	45-year-old woman with cardiovascular disease	211	N/A	Mild	Mild—home	N/A
						Mild	Mild—home	
100. Fabianova K et al. [69]—case 19	2021	Czech Republic	44-year-old woman with hypertension	169	N/A	Mild	Mild—home	N/A
						Mild	Mild—home	
101. Fabianova K et al. [69]—case 20	2021	Czech Republic	44-year-old man without comorbidities	224	N/A	Mild	Mild—home	N/A
						Mild	Mild—home	
102. Fabianova K et al. [69]—case 21	2021	Czech Republic	42-year-old woman without comorbidities	206	N/A	Mild	Mild—home	N/A
						Mild	Mild—home	
103. Fabianova K et al. [69]—case 22	2021	Czech Republic	39-year-old woman without comorbidities	229	N/A	Mild	Mild—home	N/A
						Mild	Mild—home	
104. Fabianova K et al. [69]—case 23	2021	Czech Republic	34-year-old man without comorbidities	158	N/A	Mild	Mild—home	N/A
						Mild	Mild—home	
105. Fabianova K et al. [69]—case 24	2021	Czech Republic	30-year-old woman without comorbidities	219	N/A	Mild	Mild—home	N/A
						Mild	Mild—home	
106. Fabianova K et al. [69]—case 25	2021	Czech Republic	29-year-old woman without comorbidities	139	N/A	Mild	Mild—home	N/A
						Mild	Mild—home	
107. Fabianova K et al. [69]—case 26	2021	Czech Republic	27-year-old woman without comorbidities	172	N/A	Mild	Mild—home	N/A
						Mild	Mild—home	
108. Fabianova K et al. [69]—case 27	2021	Czech Republic	27-year-old woman without comorbidities	215	N/A	Mild	Mild—home	N/A
						Mild	Mild—home	
109. Fabianova K et al. [69]—case 28	2021	Czech Republic	25-year-old man without comorbidities	222	N/A	Mild	Mild—home	N/A
						Mild	Mild—home	
110. Fernandez AC et al. [70]	2021	Portugal	28-year-old man with asthma	285	N/A	Mild	Fever, chills, sneezing	N/A
						Worse	Fever, tiredness, productive cough, frontal headache, dizziness, dark urine, dysuria	
111. Ferrante L et al. [71]	2021	Brazil	24-year-old woman without comorbidities	109	N/A	Asymptomatic	Asymptomatic	No IgG antibodies after first infection
					P1 variant	Worse	Headache, sore throat, odynophagia, nasal congestion, tiredness, fatigue, chest pain, lack of appetite, hypertension, tachycardia	
112. Fintelman-Rodrigues N et al. [72]	2021	Brazil	54-year-old man without comorbidities	65	N/A	Mild	Headache	IgM, IgA, IgG detected <1:4
					Clade 20B	Worse	Fever, dry cough, tiredness, body ache, anosmia, ageusia	IgM, IgA, IgG detected 1:128
113. Fintelman-Rodrigues N et al. [72]	2021	Brazil	57-year-old woman with discoid lupus erythematous	61	Clade 19A	Mild	Mild diarrhea	IgM, IgA, IgG detected <1:4
					Clade 20B	Worse	Fever, diarrhea, headache, body ache, anosmia, ageusia	IgM, IgA, IgG detected 1:32
114. Fintelman-Rodrigues N et al. [72]	2021	Brazil	34-year-old man without comorbidities	64	Clade 20B	Mild	Asymptomatic	IgM, IgA, IgG detected <1:4
					Clade 20B	Worse	Fever, nausea, tiredness, headache, body ache	IgM, IgA, IgG detected 1.64
115. Fintelman-Rodrigues N et al. [72]	2021	Brazil	34-year-old woman without comorbidities	60	N/A	Mild	Mild diarrhea	IgM, IgA, IgG detected <1:4
					Clade 20B	Worse	Dry cough, diarrhea, tiredness, headache, body ache, anosmia, ageusia	IgM, IgA, IgG detected 1:64
116. Fonseca V et al. [73]	2021	Brazil	29-year-old health care worker man without comorbidities	225	B.1.1.28 Spike D614G	Mild	Fever, myalgia cough, sore throat, diarrhea	IgG negative 180 days after the 1st infection
					B,1,2 Spike D614G	Mild	Again symptoms	
117. Garduno-Orbe B et al. [74]	2021	Mexico	40-year-old healthcare worker woman with hypertension, smoking	134	N/A	Moderate	Fever, dry cough, nasal drainage, dyspnea, myalgia, arthralgia, headache, anosmia, dysgeusia, decreased oxygen saturation up to 84%, maculopapular rash on the upper and lower limbs, chest, face, neck	
						Worse	Sneezing, runny nose, myalgia, arthralgia, fever, dry cough, headache, dyspnea, emphysema of the right lung	
118. Garduno-Orbe B et al. [74]	2021	Mexico	49-year-old health care worker woman with hypothyroidism	129	N/A	Mild	Nasal congestion, myalgia, arthralgia, chills, headache, dry cough, dysgeusia, anosmia, maculopapular exanthema, insomnia	
						Mild	Headache, dry cough, odynophagia, myalgia, dyspnea, conjunctivitis	
119. Garduno-Orbe B et al. [74]	2021	Mexico	53-year-old health care worker man without comorbidities	107	N/A	Mild	Fever, dyspnea, pneumonia	
						Mild	Fever, chills, anosmia, dysgeusia dry cough, rhinorrhea, general malaise, chest pain,	
120. Garduno-Orbe B et al. [74]	2021	Mexico	52-year-old health care worker man without comorbidities	82	N/A	Mild	Odynophagia, dry cough, nasopharyngeal exudate	
						Worse	Myalgias, arthralgias, dry cough, dyspnea, odynophagia, pneumonia> intensive care for hypoxia	
121. Garg J et al. [75]	2021	India	30-year-old health care worker man without comorbidities	90	N/A	Mild	Fever	30 days after initial diagnosis IgG antibody negativity
						Worse	Fever, severe myalgia, anosmia, loss of taste	30 days after reinfection diagnosis IgG antibody positivity
122. Garvey MI et al. [76]	2021	UK	92-year-old man with dementia	207	1st wave	Moderate	Pyrexia, dry cough, shortness of breath, bilateral pneumonia	
					B.1.177 (Spain variant)	Moderate	Lethargy, persistent cough, pyrexia, pneumonia	
123. Garvey MI et al. [76]	2021	UK	84-year-old man with dementia and Paget’s disease	224	1st wave	Mild	Lethargy, confusion, headache, fatigue	
					B.1.177 (Spain variant)	Mild	Positive	
124. Garvey MI et al. [76]	2021	UK	59-year-old man with end stage renal failure	236	1st wave	Mild	Cough, fluctuating temperature	
					B.1.1.7 (Kent variant)	none	None	
125. Goel N et al. [77]	2021	USA	59-year-old man with end stage renal failure and hemodialysis	59	N/A	Moderate	Cough, fever, pneumonia > hospitalization	
						Milder	Cough, shortness of breath, >hospitalization	SARS-CoV-2 IgG antibody positive after re-infection
126. Goldman JD et al. [30]	2020	USA (Washington)	Sexagenarian (age between 60 and 69) with emphysema and hypertension	140	Clade 19B	Severe	Fever, chills, productive cough, dyspnea, chest pain	
					Clade 20A harboring the spike variant D614G	Severe, but milder than first	Dyspnea, dry cough, weakness	RBD, spike and NC IgG, spike IgM, NC IgA+ on D14 of reinfection
127. Gulati K et al. [78]	2021	UK	61-year-old south Asian with immunosuppression for ANCA-associated vasculitis	180	N/A	Severe	Dry cough, dyspnea, fever, myalgia, kidney dysfunction, pneumonia	N/A
						Moderate	Fever, myalgia, dyspnea, pneumonia	
128. Gupta V et al. [79]	2020	India	25-year-old male healthcare worker	108	9 SNVs compared to initial infection (19A first infection–20A second infection)	Asymptomatic	Asymptomatic	N/A
						Asymptomatic	Asymptomatic with higher viral load	
129. Gupta V et al. [79]	2020	India	28-year-old female healthcare worker	111	10 SNVs compared to initial infection; mutation 22882T > G (S:N440K) within the receptor binding domain found in the second episode	Asymptomatic	Asymptomatic	N/A
						Asymptomatic	Asymptomatic with higher viral load	
130. Habadi MI et al. [80]	2021	SAU	44-year-old woman healthcare worker	108	N/A	Moderate	Fever, chills, severe sore throat, fatigue	N/A
						Moderate	Severe persistent productive cough, runny nose, loss of smell, partial loss of taste	
131. Habadi MI et al. [80]	2021	SAU	35-year-old heavy male smoker	94	N/A	Asymptomatic	Asymptomatic	N/A
						Worse	Fever, cough, body ache, abdominal pain, loss of taste	
132. Hanif M et al. [81]	2020	Pakistan	58-year-old cardiac surgeon male without comorbidities	55	N/A	Hospitalized for 30 days	Fatigue, headache, sore throat, pneumonia	N/A
						Hospitalized for 14 days	Fever >39 °C, headache, muscle aches	
133. Harrington D et al. [82]	2021	UK	78-year-old man with type 2 diabetes mellitus, diabetic nephropathy, chronic obstructive pulmonary diseases, sleep apnea, ischemic heart disease	250	Lineage B.2 with no mutations in the S region	Discharged home	Mild illness	SARS-CoV-2 antibodies (using the Roche anti-SARS-CoV-2 IgM/IgG assay detecting antibodies targeting viral nucleocapsid “N” antigen) were detectable on 6 occasions between 4 June 2020 and 13 November 2020 with no evidence of antibody waning seen
					Variant VOC-20201/01 of lineage B.1.1.7 with 18 amino acid replacement and deletions in the S region	Emergency intubation, worse	Shortness breath, severe hypoxia, pneumonia, myocardial infarction	
134. Hayes B et al. [83]	2021	USA	30-year-old female healthcare worker with idiopathic thrombocytopenic purpura, pancreatitis, GERD, anxiety, recurrent pneumonia	183	N/A	Mild	Fever, fatigue, sore throat, nasal congestion, dry cough, chest tightness	After 1st infection anti-SARS-CoV-2 IgG were negative
						Mild	Headaches, fever, sinus congestion	After 2nd infection anti-SARS-CoV-2 IgG were positive
135. Hunsinger HP et al. [84]	2021	USA	81-year-old woman with immunosuppression for rheumatoid arthritis	62	N/A	Mild	Altered mental status,	N/A
						Moderate	Cough, shortness of breath, oxygen requirement	
136. Hussein NR, Musa DH et al. [85]	2021	Iraq	39-year-old man with hypertension	112	N/A	Moderate	Fever, dry cough, hypoxemia	SARS-CoV-2 2 months after discharge
						Mild	Fever, not hypoxemia	
137. Hussein NR, Rashad BH et al. [86]—case 1	2021	Iraq	32-year-old man	82	N/A	Mild	Myalgia, fever	N/A
						Mild	Myalgia	
138. Hussein NR, Rashad BH et al. [86]—case 2	2021	Iraq	40-year-old man	50	N/A	Severe	Fever, loss of smell, myalgia, dyspnea	N/A
						Mild	Fever, sore throat	
139. Hussein NR, Rashad BH et al. [86]—case 3	2021	Iraq	46-year-old man	74	N/A	Mild	Fever, dry cough	N/A
						Moderate	Fever, sore throat, loss of taste and smell	
140. Hussein NR, Rashad BH et al. [86]—case 4	2021	Iraq	39-year-old man	122	N/A	Severe	Fever, dry cough, dyspnea	N/A
						Mild	Fever, sore throat	
141. Hussein NR, Rashad BH et al. [86]—case 5	2021	Iraq	32-year-old woman	174	N/A	Mild	Fever, dry cough, loss of smell, sore throat	N/A
						Mild	Fever, sore throat, myalgia	
142. Hussein NR, Rashad BH et al. [86]—case 6	2021	Iraq	44-year-old man with colon cancer	51	N/A	Mild	Fever, myalgia	N/A
						Mild	Myalgia	
143. Hussein NR, Rashad BH et al. [86]—case 7	2021	Iraq	26-year-old woman	84	N/A	Mild	Headache, sweating, loss of taste	N/A
						Mild	Headache, myalgia	
144. Hussein NR, Rashad BH et al. [86]—case 8	2021	Iraq	26-year-old woman	84	N/A	Mild	Headache, loss of taste	N/A
						Moderate	Myalgia, cough, dyspnea	
145. Hussein NR, Rashad BH et al. [86]—case 9	2021	Iraq	36-year-old woman with diabetes	51	N/A	Mild	Sore throat, fever	N/A
						Severe	Fever, myalgia, cough, dyspnea	
146. Hussein NR, Rashad BH et al. [86]—case 10	2021	Iraq	34-year-old man	49	N/A	Mild	Headache, fever	N/A
						Severe	Myalgia, fever, headache, anorexia	
147. Hussein NR, Rashad BH et al. [86]—case 11	2021	Iraq	79-year-old woman with heart failure and hypertension	58	N/A	Severe	Fever, dyspnea	N/A
						Severe	Cough, anorexia, fever	
148. Ibrahim M et al. [87]	2021	USA	59-year-old Caucasian male with Hodgkin lymphoma	150	N/A	Moderate	Shortness of breath, dry cough, tachycardia, oxygen desaturation to 85%	N/A
						Moderate	Chills, worsening shortness of breath, productive cough, fever, tachycardia, hypoxemia	
149. Inada M et al. [88]	2021	Japan	58-year-old with mild dyslipidemia	105	N/A	Moderate	Fever, bilateral pneumonia	After 1st episode IC50 of neutralizing antibodies anti-SARS-CoV-2 was 50.0 microg/mL
						Asymptomatic	Asymptomatic	After 2nd episode IC50 of neutralizing antibodies anti-SARS-CoV-2 was 14.8 microg/mL
150. Jain A et al. [89]	2020	India	21-year-old female	50	N/A	Asymptomatic	Asymptomatic	N/A
						mild	Complete loss of smell for 2 weeks	
151. Kapoor R et al. [90]	2021	India	39-year-old male with multiple myeloma	84	N/A	Asymptomatic	Asymptomatic	N/A
						Severe	High grade fever, chills, shortness of breath, bilateral pneunomia	
152. Kapoor R et al. [90]	2021	India	33-year-old male with T cell acute lymphoblastic leukemia	60	N/A	Severe	Fever, cough, pneumonia	N/A
						Severe	Headache, vomiting, high grade fever, pneumonia	
153. Kapoor R et al. [90]	2021	India	26-year-old male with Philadelphia chromosome positive acute lymphoblastic leukemia	91	N/A	Asymptomatic	Asymptomatic	N/A
						Moderate	Fever	
154. Krishna VN et al. [91]	2021	USA	70-year-old man with hypertension, diabetes mellitus, coronary artery disease	45	N/A	Asymptomatic	Asymptomatic	COVID-19 IgG positive after 1st infection
						Worse	Shortness of breath, cough, chest pain, myalgias	
155. Krishna VN et al. [91]	2021	USA	Late 50s woman with hypertension, hepatitis C, heart failure	75	N/A	Asymptomatic	Asymptomatic	N/A
						Worse	Fever, myalgias, sore throat	
156. Klein J et al. [31] *	2021	USA	66-year-old man with bipolar disorder, end-stage renal disease due to lithium toxicity and renal transplantation	210	Clade B.1	Mild	Fever, fatigue, dry cough	Failure of humoral immunity with defective response of the neutralizing antibodies after primary infection
					Clade B.1.280	Milder	Fatigue and nonproductive cough	
157. Kulkarni O et al. [92]	2021	India	61-year-old male healthcare worker	75	20B clade	Asymptomatic	Asymptomatic	N/A
					20B clade with 10 variations	Mild	Cough, weakness	
158. Larson D et al. [93]	2020	USA (Virginia)	42-year-old man military healthcare provider	64	Lineage B.1.26	Moderate, clinical resolution in 10 days	Cough, fever, myalgias	
					Lineage B.1.26 with several potential variations	Severe, worse	Fever, cough, shortness of breath, gastrointestinal symptoms, pneumonia	Spike IgG+ on D8 of reinfection
159. Lechien JR et al. [94]	2020	France	42-year-old Parisian male	7 months	N/A	Home-managed	Dyspnea, fever, headache, diarrhea, abdominal pain, ageusia, total less of smell	IgG 2 months after
						Milder	Fever, nasal burning, total loss of taste and smell	
160. Lechien JR et al. [94]	2020	Spain	38-year-old Spanish health care worker female	6 months	N/A	Moderate—hospitalized for 7 days	Dyspnea, fever, headache, diarrhea, loss of smell	N/A
						Milder	Fever, headache, new total loss of smell and taste	
161. Lee JS et al. [95]	2020	South Korea	21-year-old healthy woman	26	Clade V—found in Asia and Europe	Hospitalized with few symptoms	Sore throat	
					Clade G—found in south Korea	Mild	Cough, sore throat	IgG+
162. Loconsole D et al. [96]	2021	Italy	41-year-old healthcare worker woman	289	20B	Mild	Fever, arthralgia, headache, diarrhea, anosmia, ageusia	IgG positive after 1st infection and after 2nd infection
					20E (EU1)	Mild	Headache, sore throat, diarrhea	
163. Loh SY et al.	2021	UK	55-year-old man with X-linked agammaglobulinemia	56	N/A	Moderate	Purulent sputum, fever, breathlessness, fever, headache, myalgia, chest tightness	N/A
						Worse	Short of breath, fevers > death	
164. Luciani M et al. [97]	2020	Italy	69-year-old man, heavy smoker with classic Hodgkin’s lymphoma with mixed cellularity	131	N/A	Moderate with 3 months of hospitalization	Pneumonia, fever, diarrhea	IgG+ 50 days after hospitalization
						Moderate with 64 days of hospitalization	Fever, dyspnea, anemia, leukopenia, pneumonia	N/A
165. Mahajan NN et al. [98]—Case 2	2021	India	33-year-old man	90	N/A	Mild	Sore throat	N/A
						Worse	Influenza like Illness symptoms with breathing difficulty	
166. Mahajan NN et al. [98]—Case 3	2021	India	27-year-old man	69	N/A	Asymptomatic	Asymptomatic	N/A
						Worse	Fever, cough, myalgia	
167. Mahajan NN et al. [98]—Case 4	2021	India	48-year-old woman	97	N/A	Mild	Myalgia	N/A
						Mild	Myalgia	
168. Mahajan NN et al. [98]—Case 5	2021	India	26-year-old woman	55	N/A	Mild	Fever, myalgia	N/A
						Mild	Fever, sore throat, myalgia	
169. Mahajan NN et al. [98]—Case 6	2021	India	25-year-old man	89	N/A	Mild	Fever, sore throat, myalgia and loss of smell and taste	N/A
						Mild	Fever	
170. Mahajan NN et al. [98]—Case 7	2021	India	31-year-old man	70	N/A	Asymptomatic	Asymptomatic	N/A
						Worse	Myalgia	
171. Mahajan NN et al. [98]—Case 9	2021	India	51-year-old woman	157	N/A	Asymptomatic	Asymptomatic	N/A
						Worse	Myalgia, headache, pneumonia (25% lung involvement)	
172. Marquez L et al. [99]	2021	USA	16-year-old woman with end-stage renal disease	90	B.1.2	Mild	Sore throat, fatigue, nasal congestion, rhinorrhea, dry cough	IgM+ and IgG− after the 2nd infection
					B.1.1.7	Milder	Leg pain, fatigue, swelling leg, fever	
173. Massanella M et al. [100]	2021	Spain	62-year-old male healthcare worker with previous history of mild asthma, hypertension, dyslipidemia, liver steatosis, hyperuricemia, and overweight (body mass index ≥ 30 kg/m^2^)	158		Mild	Fever of 38 °C, diarrhea, anosmia, dysgeusia, cough, intense asthenia, and arthromyalgia	After reinfection weak immune response, with marginal humoral and specific T-cell responses against SARS-CoV-2. All antibody isotypes tested as well as SARS-CoV-2 neutralizing antibodies increased sharply after day 8 post symptoms. A slight increase of T-cell responses was observed at day 19 after symptom onset
					B.1.79 (G)	Worse	Intense arthromyalgias, headache, fever, cough, and dyspnea > admitted to the emergency room for worsening dyspnea, cough, chills, fever 39 °C, myalgias, anosmia, and ageusia. His respiratory rate was 36 breaths/minute, his heart rate was 100 beats/minute, and he had bilateral inspiratory crackles. The chest radiograph showed bilateral alveolar-interstitial infiltrates	
174. Mohseni M et al. [101]	2021	USA	53-year-old female with liver transplant in 2010 due to alcoholic cirrhosis, hypertension, hypothyroidism, anxiety, and chronic kidney disease	90	N/A	Severe	Encephalopathy due to her COVID-19	N/A
						Mild	Nausea, vomiting, diarrhea, and myalgias	
175. Mulder et al. [102]	2020	Denmark	89-year-old immunocompromised woman (Waldestrom macroglobulinemia)	59	The 2 strains differed at 10 nucleotide positions in ORF1a (4), ORF1b (2), spike (2), ORF3A (1), M (1) genes	Hospitalized for 5 days	Fever, severe cough, persisting fatigue	IgM-
						Worse	Fever, cough, dyspnea > death after 2 weeks	N/A
176. Munos Mendoza J et al. [103]	2020	USA	51-year-old African American male with hypertension and hemodialysis history	2 months	N/A	Asymptomatic	Positive for NAAT and IgG at a routine control during hemodialysis	IgM−, IgG+
						Severe, hospitalized with non-invasive positive pressure mechanical ventilation	Fever 38.3 °C, severe dyspnea, pneumonia	IgG+, IgM+, IgA+
177. Nachmias V. et al. [104]	2020	Israel	22-year-old woman without comorbidities	111	N/A	Mild with home back after 23 days	Fever, cough	
						Asymptomatic	Tachycardia	IgG+
178. Naveca F et al.—case 1 [105] *	2021	Brazil	29-year-old	281	20A	Mild	Fever, myalgia, cough, sore throat, nausea, and back pain	
					20J (P.1)	Mild	Fever, cough, sore throat, diarrhea, anosmia, ageusia, headache, runny nose, and resting pulse oximetry of 97%	
179. Naveca F et al.—case 2 [105] *	2021	Brazil	50-year-old	153	20B	Mild	Fever, cough, and tiredness	
					20J (P.1)	Mild	Cough, headache, and runny nose	
180. Naveca F et al.—case 3 [105] *	2021	Brazil	40-year-old woman	282	20A	Mild	Fever, headache, chest pain, and weakness	
					20J (P.1)	Mild	Sore throat and running nose	
181. Nazar N et al. [106]	2020	India	26-year-old man healthcare worker	97	N/A	Asymptomatic	Asymptomatic	N/A
						Asymptomatic	Asymptomatic	
182. Nicholson EG et al. [107]—case 1	2021	USA	46-year-old man with hypertension, gastroesophageal reflux disease, plantar fasciitis	>90	N/A	Mild	Fever, myalgias, sore throat, chills, headaches, nausea, shortness of breath	SARS-CoV−2 IgG testing 1st test: 1:4096 (BCM laboratory)
						Asymptomatic	Asymptomatic	SARS-CoV−2 IgG testing 2nd test: 1:2048 (BCM laboratory)
183. Nicholson EG et al. [107]—case 2	2021	USA	27-year-old woman	>90	N/A	Mild	Congestion, fatigue, loss of taste, loss of smell, headache	N/A
						Milder	Fever, chills, fatigue	
184. Nicholson EG et al. [107]—case 3	2021	USA	53-year-old man with hypertension, sleep apnea	>90	N/A	Mild	Cough, congestion, loss of taste, loss of smell	SARS-CoV−2 IgG testing 1st test: 1:2048 (BCM laboratory)
						Asymptomatic	Asymptomatic	SARS-CoV−2 IgG testing 2nd test: 1:1024 (BCM laboratory)
185. Nicholson EG et al. [107]—case 4	2021	USA	66-year-old woman with diabetes mellitus, rheumatoid arthritis, systemic lupus erythematosus, congestive heart failure, renal disease, gout, hypertension	>90	N/A	Mild	Fatigue	N/A
						Asymptomatic	Asymptomatic	
186. Nicholson EG et al. [107]—case 5	2021	USA	73-year-old woman with hypertension, hyperlipidemia, depression	>90	N/A	Mild	Congestion, sore throat, headache	N/A
						Mild	Cough, shortness of breath, congestion, abdominal pain, nausea, vomiting, headache	
187. Nicholson EG et al. [107]—case 6	2021	USA	42-year-old woman with breast cancer	>90	N/A	Mild	Cough, shortness of breath, fatigue, loss of taste, loss of smell, headache, fever	SARS-CoV−2 IgG testing 1st test: 1:4096 (BCM laboratory)
						Asymptomatic	Asymptomatic	
188. Nicholson EG et al. [107]—case 7	2021	USA	36-year-old man	>90	N/A	Mild	Cough, fatigue, nausea, loss of smell, fever	SARS-CoV−2 IgG testing 1st test: 1:4096 (BCM laboratory), 2nd test: 1:4096 (BCM laboratory)
						Asymptomatic	Asymptomatic	
189. Nonaka CKV et al. [108]	2021	Brazil	45-year-old woman	147	Lineage B.1.1.33 with S:G1219C mutation	Mild	Diarrhea, myalgia, asthenia, odynophagia for 7 days	N/A
					Lineage P.2 (or B.1.1.28.2) with S:E484K mutation	Moderate	Headache, malaise, ageusia, muscle fatigue, insomnia, mild dyspnea, shortness of breath	
190. Novoa W et al. [109]	2021	Colombia	44-year-old male, healthcare worker	103	N/A	Asymptomatic	Asymptomatic	N/A
						Moderate	Malaise, chills, headache, fever, odynophagia	
191. Ozaras R et al. [110]	2020	Turkey	23-year-old woman	116	N/A	Hospitalized	Fever >39 °C, chills, fatigue, cough, headache, sore throat, muscle and joint pain	N/A
						Recovered in 10 days	Fever 28.7 °C, chills, fatigue, loss of appetite, taste and smell loss, muscle and joint pain	IgG slightly positive
192. Pow T et al. [111]	2021	USA	40-year-old man	89	N/A	Mild	Fever, cough	N/A
						Worse	Dyspnea, tachycardia > death	
193. Prado-Vivar B et al. [28]	2020	Ecuador	46-year-old man	63	Nextstrain 20A/GISAID B1.p9 lineage	Mild	Intense headache, drowsiness	IgM+ IgG− on D7 of initial infection
					Nextstrain 19B/GISAID A.1.1 lineage; 18 mutations difference	Moderate	Odynophagia, nasal congestion, fever 39 °C, back pain, productive cough, dyspnea	IgM+ IgG+ on D28
194. Quiroga B et al. [112]	2021	Spain	60-year-old male, with chronic kidney disease (CKD) due to focal and segmental glomerulosclerosis that received his first kidney transplant 2004	149	N/A	Mild	Cough and low-grade fever	Antibodies (IgM and IgG) for SARS-CoV2 resulted negative after reinfection
						Worse	Respiratory fever and acute injury of the allograft function. A chest X-ray showed bilateral infiltrates with unilateral pleural effusion > death	
195. Ramirez JD et al. [113]—case 3	2021	Colombia	54-year-old woman with hypertension, gastritis, arthrosis	33	B.1	Mild	Fever, cough, odynophagia, fatigue	N/A
					B.1.1.269	Milder	Fever, odynophagia	
196. Rani PR et al. [114]	2021	India	47-year-old man	46	15 genetic variants with 22882T > G (Spike N440K)	Asymptomatic	Asymptomatic	N/A
					17 genetic variants with 22882T > G (Spike N440K)	Worse	Fever, cough, malaise	
197. Resende PC et al. [115]	2021	Brazil	37-year-old healthcare worker woman	116	B.1.1.33	Mild	Headache, runny nose, diarrhea, myalgia	IgG+ after re-infection
					VOI P.2 with mutation S-E484K	Mild	Headache, ageusia, anosmia, fatigue	
198. Rodríguez-Espinosa D et al. [116]	2021	Spain	76-year-old man with hypertension, biological aortic heart valve replacement, and end-stage kidney disease secondary to autosomal dominant polycystic kidney disease	58		Asymptomatic	Asymptomatic	IgG and IgM to SARS-CoV-2 tested negative after 1st and 2nd episode
						Worse	Fever, cough, and shortness of breath, bilateral pneumonia > death 18 days after admission	
199. Romano CM et al. [117]	2021	Brazil	26-year-old woman	128	Non-VOC virus	Mild	Dry cough, dizziness, headache, fatigue, stuffy nose, back pain, loss of taste, nausea, diarrhea	
					VOC-virus P.1 variant	Mild	Dry cough, dizziness, headache, fatigue, diarrhea, joint pain legs, difficult breathing	
200. Salcin S et al. [118]	2021	USA	62-year-old woman with hypertension, hypothyroidism, chronic lower back pain	90	N/A	Hospitalized	Worsening shortness of breath, cough, hypoxia	N/A
						Worse with intubation twice	Tachypnea, hypoxia, pneumonia	
201. Salehi-Vaziri M et al. [119]	2021	Iran	42-year-old man	128	20G with 11 mutations	Mild	Cough, headache, severe diarrhea	IgG and IgM negative
					20G with 17 mutations	Mild	Body pain, shortness of breath, headache, anosmia	IgG and IgM negative
202. Salehi-Vaziri M et al. [120]	2021	Iran	32-year-old woman	63	N/A	Mild	Headache, sore throat, cough, fever	The antibody titration was achieved positive by the rapid test (sensitivity 72%, specificity: 76%) for IgM (At the time of second infection, IgG titration was assessed as 4.89 AU/mL which after two months turned to a significant raise (over ELISA reader standard range).
					D614G mutation	Worse	Severe cough, fever, fatigue	
203. Salehi-Vaziri M et al. [120]	2021	Iran	54-year-old man	156	L139L non-synonymous mutation	Mild	Fatigue, anxiety, chest pain, cough, fever	IgM and IgG were detected in the first incidence, and he was being followed up to the second virus presentation. In the whole duration between two incidences, IgG test was positive. Antibody titration at the time of second infection showed that IgG level was 5.25 IU/mL which increased to 27.5 IU/mL after about 2 weeks.
					L139L non-synonymous mutation	Mild	Milder fatigue, chest pain, dizziness, diarrhea	
204. Salehi-Vaziri M et al. [120]	2021	Iran	42-year-old man	111	N/A	Mild	Shortness of breath, sore throat, shaking chills, pain, diarrhea	The IgG titration was 17.5 IU/mL which decreased to 6.5 IU/mL after almost 2 weeks.
					D614G mutation	Mild	Similar to the first infection with severe diarrhea	
205. Salzer HJF [121]	2021	Austria	95-year old man with dementia, hypertension, total thyroidectomy	124	N/A	Mild	Fever, leukopenia	N/A
						Severe	Pneumonia	
206. Sanyang B et al. [122]	2021	Gambia	31-year-old woman without comorbidities	145	B1	Mild	Mild	
					B1.1.74	Mild	Mild	
207. Sanyang B et al. [122]	2021	Gambia	36-year-old woman without comorbidities	184	B.1.235	Asymptomatic	Asymptomatic	
					B.1	Worse	Mild	
208. Scarpati G et al. [123]	2021	Italy	63-year-old healthcare man with type II diabetes, atrial fibrillation, chronic obstructive pulmonary disease	299	Clade 20A	Asymptomatic	Asymptomatic	
					Clade 20E	Worse	Shortness of breath with rapid worsening of clinical presentation and recovering in intensive care unit > death	
209. Selhorst P et al. [124]	2020	Belgium	39-year-old female immunocompetent healthcare worker	185	Different clades: 19A	Mild	Cough, dyspnea, headache, fever, general malaise	IgG+
					20A	Milder	Dyspnea	IgM and IgG+
210. Selvaraj V. et al. [125]	2020	USA	70-year-old male with obesity, neuropathy, asthma, obstructive sleep apnea, hypertension	7 months	N/A	Hospitalized	Worsening shortness of breath, tachypneic, mild, patchy mid and lower lung airspace disease bilaterally	SARS-CoV-2 IgG−
						Hospitalized	Shortness of breath, fever, body aches, nausea, malaise	
211. Sen MK et al. [126]	2020	India	78-year-old man with coronary artery disease	57	N/A	Mild	Fever, cough for 2 days	N/A
						Mild	Fever, cough, dyspnea for 1 day	
212. Sevillano G et al. [127]	2021	Ecuador	28-year-old man	102	B.1.1	Mild	Sore throat, cough, headache, nausea, diarrhea, anxiety, panic attack	IgM and IgG negative after 1st infection
					Different in 27 nucleotides	Mild	Anosmia, ageusia, fever, headache	IgM and IgG negative after 2nd infection
213. Sharma R et al. [13]	2020	Qatar	57-year-old male with diabetes mellitus	86	N/A	Asymptomatic	Asymptomatic, screening for exposition to an infected work colleague	N/A
						Symptomatic	Fever, myalgia, headache, productive cough	IgM and IgG+
214. Shastri J et al. [128]—Case A	2021	India	27-year-old male doctor	66	Lineage B.1	Mild, 2 days of symptoms	Sore throat, nasal congestion, rhinitis	N/A
					Lineage B with 7 differences	Mild, worse than initial (1 week)	Myalgia, fever, non-productive cough, fatigue	Abbott anti-NC IgG− on D5 of reinfection
215. Shastri J et al. [128]—Case B	2021	India	31-year-old male doctor	65	Lineage B.1.1	Asymptomatic	Nothing	N/A
					Lineage B.1.1 with 8SPSs in initial strain compared to reference not present in reinfection strain including D614G	Mild, worse than initial (2 days)	Myalgia, malaise	Abbott NC IgG− on D7 of reinfection
216. Shastri J et al. [128]—Case C	2021	India	27-year-old male doctor	19	Lineage B.1.1	Asymptomatic	Asymptomatic—screening prior going home to visit parents	N/A
					Lineage B.1.1 with 9 SNPs compared to reference not present in initial infection strain including D614G	Mild	Fever, headache, myalgia not productive cough	IgG/IgM/IgA−
217. Shastri J et al. [128]—Case D	2021	India	24-year-old woman nurse	55	Lineage B.1.1	Mild, 5 days	Sore throat, rhinitis, myalgia	N/A
					Lineage B.1.1 with 10SNPs compared to reference not present in initial infection strain including D614G	Mild, worse than initial—3 weeks	Fever, myalgia, rhinitis, sore throat, not productive cough, fatigue	IgG/IgM/IgA−
218. Shoar S et al. [129]	2021	USA	31-year-old healthcare worker man	79	N/A	Severe	Malaise, cough, shortness of breath, anosmia, =2 saturation to 88%, pneumonia	N/A
						Milder	Malaise, aphthous gingival ulcer, desquamating palmar lesion, fever, myalgia	
219. Sicsic I et al. [130]	2021	USA	69-year-old woman with asthma, hypercholesteremia, hypertension, OSA (obstructive sleep apnea)	70	N/A	Mild	Shortness of breath, dry cough, headache, fatigue, fevers	N/A
						Moderate	Cough, fever, ageusia	
220. Siqueira JD et al. [131]	2021	Brazil	76-year-old woman with chronic renal failure and renal squamous cell carcinoma	104	9 single nucleotide variations (SNVs)	Severe	Cough, fever, pneumonia	N/A
						Worse	Cough, fever, pneumonia > death	
221. Soares da Silva et al. [132]	2021	Brazil	39-year-old man with chronic cardiovascular disease, diabetes mellitus	101	P.1	Not reported	Not reported	N/A
					P.2	Worse	Dyspnea, fatigue, respiratory distress > intubated > death 12 days after the onset of symptoms	
222. Staub T et al. [133] – case 1	2021	France	Mid-20s healthcare worker man without comorbidities	>83	November N/A	Asymptomatic	Asymptomatic	N/A
					B1.351—identified in December 2020 in South Africa	Worse	Cough	
223. Staub T et al. [133]—case 2	2021	France	Mid-20s healthcare worker woman without comorbidities	288	April 2020—N/A	Mild	Fever, headache, chills, diarrhea, loss of taste and smell	N/A
					B1.351	Milder	Fever, headache, chills	
224. Staub T et al. [133]—case 4	2021	France	Late-20s healthcare worker woman without comorbidities	90	November 2020—N/A	Mild	Fever, muscle pain, headache, loss of taste and smell	N/A
					B1.351	Milder	Cough, muscle pain	
225. Takeda C et al. [134] Patient 1	2020	Brazil	29-year-old man healthcare professional without comorbidities	53	N/A	Mild	Myalgia, fever	N/A
						Mild	Fever, anosmia, loss of taste	
226. Takeda C et al. [134] Patient 2	2020	Brazil	63-year-old man healthcare professional without comorbidities	58	N/A	Mild	Diarrhea, fever	N/A
						Mild	Hypoxemia, fever	
227. Takeda C et al. [134] Patient 3	2020	Brazil	40-year-old woman healthcare professional with ankylosing spondylitis and asthma	70	N/A	Moderate	Fever, Pneumonia, myalgia	Not specified
						Mild	Anosmia, fever	
228. Takeda C et al. [134] Patient 4	2020	Brazil	67-year-old man healthcare professional with obesity, apnea syndrome, rhinitis	54	N/A	Mild	Coryza, arthralgia	Not specified
						Hospitalized with high-flow oxygen therapy	Hypoxia	
229. Takeda C et al. [134] Patient 5	2020	Brazil	47-year-old man healthcare professional without comorbidities	56	N/A	Mild	Myalgia, fever	Not specified
						Mild	Fever	
230. Takeda C et al. [134] Patient 6	2020	Brazil	31-year-old man healthcare professional without comorbidities	57	N/A	Moderate	Hypoxemia, myalgia, diarrhea, fever	Not specified
						Moderate	Hypoxemia, fever	
231. Tang CY et al. [135]	2021	USA	Female in 20s with asthma, obesity, anxiety, depression	19	PANGOLIN A.3 lineage	Mild	Cough, chills, exertional dyspnea, sore throat, dizziness, rhinorrhea, fever	N/A
					PANGOLIN B.1.1 lineage	Milder		
							Cough, fatigue, dyspnea	
232. Tehrani HA et al. [136]	2021	Iran	15-year-old boy with acute myeloid leukemia M3	43	N/A	Moderate	Cough, dyspnea, patchy infiltration in the left lung	IgG+ IgM−
						Severe	Fever, neutropenia, cough, myalgia and shivering, O2 saturation at 75%, pneumonia	IgG−
233. Teka IA et al. [137]	2021	Libya	18-year-old man	80	N/A	Mild	Fever, headache, sore throat, cough, shortness of breath, anosmia	IgG positive after re-infection
						Worse	Fever, cough, muscle pain, dyspnea, hypoxia	
234. Tillett RL et al. [27]	2020	USA (Nevada)	25-year-old man without comorbidities	48	Clade 20C	Mild	Sore throat, cough, headache, nausea, diarrhea	N/A
					Clade 20C with 11SNP mutation	Severe with hospitalization	Fever, headache, dizziness, cough, nausea, diarrhea, hypoxia, shortness of breath	Roche Elecsys Anrti-SARS-CoV-2 IgM/IgG+ on D8 of reinfection
235. To KK et al. [5,138,139]	2020	Hong Kong	A 33-year-old male	142	Nextstrain 19A/GISAID V/Pangolin lineage B.2	Mild—hospitalized	Fever, headache, cough, sore throat	IgG negativity by ELISA or microsphere based antibody assay 10 days post symptom onset; IgG positivity but IgM negativity by indirect immunofluoresence assay; neutralizing antibody presence 10 days post-symptom onset with conventional and pseudovirus-based neutralization tests (VNTs)
					Nextstrain 20A/GISAID G/Rambout B.1.79; 24 nucleotides difference	Asymptomatic, systematic screening	Asymptomatic	IgG negativity by ELISA or microsphere based antibody assay 1 day post-hospitalization, but positivity at day 5; absence of neutralizing antibodies by VNTs and IgM negativity by IFI assay and CLIA 1 day post-hospitalization; then positivation on day 3; neutralizing antibody detection on day 3; IgG detection by IFon day 3; high affinity IgG
236. Tomkins-Tinch C-H et al. [140]	2021	USA	61-year-old man with liver transplant due to chronic hepatitis B and C infections	111	Genome of 2nd episode differed by 11 to 12 single base substitutions	Mild	Fever, nausea, vomiting, cough	
						Worse	Confusion, hallucination, lethargy, hypoxia	Anti-SARS-CoV-2 assay positive after 2nd episode
237. Tomassini S et al. case 9 [141]	2021	UK	93-year-old British male with multiple myeloma, cognitive impairment	55	N/A	14 days—hospitalized	Lethargy, reduced appetite, diarrhea	
							Cough, fever, dyspnea	Abbott Architect SARS-CoV-2 IgG+ on D58
238. Tomassini S et al. case 24 [141]	2021	UK	82-year-old British male with atrial fibrillation, congestive cardiac failure, abdominal aortic aneurism, lung cancer, diabetes	87	N/A	Mild—hospitalized	Fever, cough, sore throat, dyspnea, hemoptysis, hypoxia	
						Milder	Fever, cough, dyspnea	Abbott Architect SARS-CoV-2 IgG+ on D88. 92
239. Torres DA et al. [142]	2020	Brazil	36-year-old female medical doctor without comorbidities	87	N/A	Moderate	Rhinorrhea, sore throat, low fever, diarrhea, asthenia, mild headache, erythematous vesicles on her right calf, severe musculoskeletal pain of the lower limbs, hyperesthesia	IgG− 23 days after the onset, IgM/IgG− after 33 and 67 days from onset
						Worse	Nasal obstruction, hyaline rhinorrhea, sudden and complete anosmia and ageusia, frontal headache and asthenia, pneumonia	IgG+ at the 20th day
240. Tuan J et al. [136]	2021	USA	44-year-old Hispanic man with type 2 diabetes mellitus, obesity	4 months	N/A	Severe with tracheostomy	Dyspnea, stridor, difficulty at breath,	IgG+
						Mild	Fever, respiratory decompensation	
241. Ul-Haq Z et al. [143]	2020	Pakistan	41-year-old healthcare worker man	133	N/A	Mild	Fever, oxygen saturation of 90–92%, bilateral lung infiltrates, mild shortness of breath, loss of taste, severe restlessness, insomnia, body-aches	SARS-CoV-2 antibodies: 1.97
						Milder	Fever, moderate shortness of breath, loss of smell, moderate restlessness, insomnia, body aches	SARS-CoV-2 antibodies: 0.08
242. Van Elsland J et al. [29]	2020	Belgium	51-year-old woman with asthma	93	Pangolin Lineage B.1.1	Moderate with self-quarantine for 2 weeks	Headache, myalgia, fever, cough, chest pain, dyspnea; some persisting symptoms for 5 weeks	N/A
					Lineage A; 11 nucleotide differences	Milder with resolution in 1 week	Headache, cough, fatigue, rhinitis	Roche nucleocapsid IgG+ on D7 of reinfection
243. Vetter P et al. [144]	2021	Switzerland	36-year-old female physician	205	Clade 20A	Mild	Asthenia, headache, slight memory loss	Positivity for anti-S1 IgG and anti-N Ig at 14th and at 30th days
					Clade 20A.EU2 with non-synonymous mutation in the S (S477N)	Mild	Asthenia, shivering, rhinorrhea, anosmia, arthralgia, headache, exertional dyspnea for 10 days	Positivity for anti-S1 IgG and anti-N Ig
244. Vora T et al. [145]	2021	India	58-year-old woman with hypertension, hypothyroidism	120	N/A	Mild	Fever, generalized body ache, running nose, soreness of throat	Total antibody and immunoglobulin G antibody test for COVID-19 were negative after first infection
						Milder	Fever, generalized body ache, dry cough, throat pain	
245. Vora T et al. [145]	2021	India	58-year-old woman with hypertension and hypothyroidism	91	N/A	Mild	Low-grade intermittent fever, generalized body ache, running nose and soreness of throat	N/A
						Mild	Intermittent fever, generalized body ache, dry cough, and throat pain	
246. West J et al. [146]	2021	UK	25-year-old male UK doctor	17	N/A	Mild	High-grade fevers, headache of 3-day duration, severe fatigue lasting 3 weeks	N/A
						Milder	Fatigue, coryzal symptoms for 4 days	Rest at home
247. Yeleti R et al. [147]	2021	USA	25-year-old female medical student with vitiligo	120	N/A	Asymptomatic	Asymptomatic	IgG+
						Severe	Fever, abdominal pain, fatigue, vomiting and fulminant myocarditis with co-infection of parvovirus and SARS-CoV-2	N/A
248. Yu ALF et al. [148]	2021	Brazil	41-year-old woman with gastroplasty history	146	B.1.1.33 lineage	Mild	Headache, myalgia, fatigue, fever, dry cough, shortness of breath, anosmia, loss of taste	N/A
					B.1.1.28 lineage	Mild	Headache, myalgia, fatigue, fever, dry cough, shortness of breath, anosmia, loss of taste, diarrhea, loss of appetite, dizziness	
249. Yu ALF et al. [148]	2021	Brazil	34-year-old healthcare worker woman with chronic respiratory disease	173	B.1.1.28 lineage	Mild	Fever, cough, odynophagia, dyspnea	N/A
					P2	Mild	Headache, running nose, fever, sore throat	
250. Zare F et al. [149]	2021	Iran	50-year-old man	230	N/A	N/A	N/A	N/A
251. Zare F et al. [149]	2021	Iran	81-year-old woman	234	N/A	Moderate	N/A	N/A
						Worse	Death for COVID-19	
252. Zare F et al. [149]	2021	Iran	42-year-old woman	107	N/A	N/A	N/A	N/A
253. Zare F et al. [149]	2021	Iran	27-year-old man	115	N/A	N/A	N/A	N/A
254. Zare F et al. [149]	2021	Iran	79-year-old man	150	N/A	Moderate	N/A	N/A
						Worse	Death for COVID-19	
255. Zare F et al. [149]	2021	Iran	86-year-old man	164	N/A	Moderate	N/A	N/A
						Worse	Death for COVID-19	
256. Zare F et al. [149]	2021	Iran	90-year-old woman	130	N/A	N/A	N/A	N/A
257. Zare F et al. [149]	2021	Iran	13-year-old woman	124	N/A	N/A	N/A	N/A
258. Zhang K et al. [150]	2020	China	33-year-old female	59	N/A	Moderate—hospitalized for 16 days		Reduction of IgG+ to −
						Moderate		IgG+
259. Zhang K et al. [150]	2020	China	33-year-old female	86	N/A	Severe—hospitalized for 38 days		Reduction of IgG+ to weak+
						Moderate		IgM+ and IgG+
260. Zucman N et al. [151]	2021	South African	58-year-old male with asthma	120	N/A	Mild	Dyspnea, fever	IgG+
					South African variant 501Y.V2	Severe with intubation and mechanical ventilation	Dyspnea, fever, severe acute respiratory distress syndrome	

* data from papers which are not certified by peer review, medRxiv or Research Square preprints. ** days from recovery not from 1 infection. NAAT: nasopharyngeal nucleic acid amplification test; AT: antibody test.

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
