# Peer review of "SARS-CoV-2 Reinfection Is a New Challenge for the Effectiveness of Global Vaccination Campaign: A Systematic Review of Cases Reported in Literature"

_ijerph, 2021, doi:10.3390/ijerph182011001_

Round 1

Reviewer 1 Report

This manuscript conducted a systematic search of literature on SARS-CoV-2 reinfection. There were 118 articles selected with 264 confirmed cases in total. Many of the reinfection cases were less severe than the initial cases. The different clades or lineages between reinfection and initial infection were detected. The authors concluded that the findings are useful and contribute towards the role of vaccination in response to reinfection.

Comments:

  • Why do the authors list all the 264 confirmed cases in the manuscript? It is unnecessary. In addition, they describe the demographic and clinical features of the reinfection cases only in the text. They should summarize these data in concise tables. In this way, the data could be shown in a more clear and concise manner.

  • The sentences from lines 215 to 221 are completely the same as the ones from lines 225 to 230.

  • The objective of this study is to carry out a systematic search of literature on SARS-CoV-2 reinfection in order to understand the success of the global vaccine campaigns. However, the authors have not demonstrated clearly and adequately on how SARS-CoV-2 reinfection is a new challenge for the effectiveness of the global vaccination campaigns. Only a few references that described reinfection after vaccination were included.

  • There are many grammar, wording, vocabulary and spelling errors. There are English errors in every paragraph. This manuscript seems to have been barely checked before submission.

  • The format of the tables is inconsistent. In addition, it is suggested that the authors use the international format for numbering.

Author Response

  • Why do the authors list all the 264 confirmed cases in the manuscript? It is unnecessary. In addition, they describe the demographic and clinical features of the reinfection cases only in the text. They should summarize these data in concise tables. In this way, the data could be shown in a more clear and concise manner. reply: We reported all 264 confirmed cases in order to allow to other researchers to see the selected cases (some cases reported in serial cases have been excluded according to exclusion criteria). We added some graphics according to 2nd reviewer’s advice

  • The sentences from lines 215 to 221 are completely the same as the ones from lines 225 to 230. Reply: we deleted Lines 215-221: The sentences from "Although reinfection" to "immune escape"
  • The objective of this study is to carry out a systematic search of literature on SARS-CoV-2 reinfection in order to understand the success of the global vaccine campaigns. However, the authors have not demonstrated clearly and adequately on how SARS-CoV-2 reinfection is a new challenge for the effectiveness of the global vaccination campaigns. Only a few references that described reinfection after vaccination were included. Reply: We added “Recently breakthrough infections were reported following mRNA vaccination in healthy subjects(5, 6), despite evidence of effective immune response among the breakthrough subjects(7). Another study reported 8 symptomatic SARS-CoV-2 infection occurred in fully vaccinated health care workers (incidence rate 4.7 per 100 000 person-days adjusted)(4).”

  • There are many grammar, wording, vocabulary and spelling errors. There are English errors in every paragraph. This manuscript seems to have been barely checked before submission. Reply: We revised English

  • The format of the tables is inconsistent. In addition, it is suggested that the authors use the international format for numbering. Reply: We used journal format

Reviewer 2 Report

Lines 16-19: For the sake of clarity, please rephrase sentences from "a systematic search" to "confirmed recover".
Line 43: Insert a hyphen between "Cov" and "2" in "SARS-CoV2".
Line 45: Add "be" before "due".
Line 61: Put "vessels" in the singular.
Line 84: Add "days" after "45".
Line 86: Put "problems" in the singular.
Line 89: Change comma to decimal point in "0,9%".
Line 97: Remove the extra space after "COVID-19".
Line 101: Insert a space between "95%" and "CI" in "95%CI". 
Line 172: Add a period to the end of the legend in Figure 1.
Line 176: Why are data from papers not certified by peer review (e.g., medRxiv preprint server) shown if authors determined that "original papers that were peer-reviewed and published in English and fulfilled the eligibility criteria were included in the final report" (lines 126-128)?
Line 179: Quantitative data throughout this section could be expressed as graphs for better visualization.
Line 198: Put "case" in the plural.
Lines 209-210: The sentence from "The observation" to "partial protection from disease" is a repetition of the sentence in lines 200-202.
Lines 225-230: The sentences from "Although reinfection" to "immune escape" are repetitions of the sentences in lines 215-221.
Lines 237-239: This paragraph contains no information on the data collected by the authors, but a general instruction for writing a certain section of the manuscript.
Lines 247-248: Add "to collection of" after "submitted" and put "samples" in the singular.
Line 253: Change "is" to "was".
Lines 258/260: Please check the format of Ct values.
Line 265: Remove the last "r" in "worser".
Line 269: Change "is" to "are" and add "a" before "different".
Lines 270-271: Correct the question as follows: "are all cases reported in the literature as reinfection by SARS-CoV-2 true reinfections?"
Lines 272/286: Replace the last comma with "and".
Line 289: Replace the first comma with "and".
Line 296: As viruses are not cultivable agents, change "virus culture" to "virus isolation".
Line 303: Remove the extra space after "lasting".
Line 304: Change "for" to "due to".
Line 308: Remove the lowercase "r" in "rRT-PCR".
Line 317: In Figure 2, remove the red underlines and change "viral culture" to "virus isolation".
Line 323: With "unwanted host", did the authors mean "unwanted virus"?
Line 327: Remove the extra space before "However".
Line 330: Change "virus" to "virion".
Line 339: Remove "continue to" before "remain".
Lines 352-353: Please consider that anti-SARS-CoV-2 IgM antibodies were already shown to persist for up to 8 months post-COVID-19 (DOI: 10.4236/crcm.2021.109029).
Line 371: Replace the last comma with "and".
Line 393: COVID-19 is the disease, not the etiological agent.
Line 404: Put the first letter in "Anti-SARS-CoV-2" in lowercase.
Line 414: Remove the extra space before "mutations".
Line 422: Replace the comma after "cytomegalovirus" with "and".
Line 423: Reverse the word order in "diagnosis differential".
Line 436: Insert a hyphen between "antibody" and "dependent".
Line 443: The antibody-dependent enhancement phenomenon had already been named in line 436.
Line 453: Put the second "s" in "with SARs-CoV-2" in upper case and insert a hyphen between "Cov" and "2" in "SARS-CoV2 RNA".
Line 480: Add "-19" after "COVID" in "COVID episode".
Line 506: Add "by" after "reported".
Line 507: Put "recognize" in the gerund.
Line 530: Change "form" to "from" and remove the extra space after "literature".
Line 542: Put "covid-19" in uppercase and remove "as seen sometimes" after "since".

Author Response

Lines 16-19: For the sake of clarity, please rephrase sentences from "a systematic search" to "confirmed recover".
we rephrased

Line 43: Insert a hyphen between "Cov" and "2" in "SARS-CoV2".
we inserted a hyphen between "Cov" and "2" in "SARS-CoV2".

Line 45: Add "be" before "due".

We added “be” before “due”

Line 61: Put "vessels" in the singular.

We modified "vessels" in the singular.

Line 84: Add "days" after "45".

We added "days" after "45".

Line 86: Put "problems" in the singular.

We modified  "problems" in the singular.

Line 89: Change comma to decimal point in "0,9%".

We changed comma to decimal point in "0,9%".

Line 97: Remove the extra space after "COVID-19".
we removed the extra space after "COVID-19"

Line 101: Insert a space between "95%" and "CI" in "95%CI".

We Inserted a space between "95%" and "CI" in "95%CI".

Line 172: Add a period to the end of the legend in Figure 1.
we added a period to the end of the legend in Figure 1.

Line 176: Why are data from papers not certified by peer review (e.g., medRxiv preprint server) shown if authors determined that "original papers that were peer-reviewed and published in English and fulfilled the eligibility criteria were included in the final report" (lines 126-128)?
we modified the inclusion criteria and substituted 2 papers that are been reviewed

Line 179: Quantitative data throughout this section could be expressed as graphs for better visualization.

We added some graphics

Line 198: Put "case" in the plural.

We modified "case" in the plural.

Lines 209-210: The sentence from "The observation" to "partial protection from disease" is a repetition of the sentence in lines 200-202.

We deleted the sentence from "The observation" to "partial protection from disease" in lines 200-202.

Lines 225-230: The sentences from "Although reinfection" to "immune escape" are repetitions of the sentences in lines 215-221.
we deleted Lines 215-221: The sentences from "Although reinfection" to "immune escape"

Lines 237-239: This paragraph contains no information on the data collected by the authors, but a general instruction for writing a certain section of the manuscript.

We deleted lines 237-9

Lines 247-248: Add "to collection of" after "submitted" and put "samples" in the singular.

We added "to collection of" after "submitted" and modified "samples" in the singular

Line 253: Change "is" to "was".

We changed “is” to “was”

Lines 258/260: Please check the format of Ct values.

the format of Ct values is correct.

Line 265: Remove the last "r" in "worser".

We removed the last "r" in "worser".

Line 269: Change "is" to "are" and add "a" before "different".

We hanged "is" to "are" and added "a" before "different".

Lines 270-271: Correct the question as follows: "are all cases reported in the literature as reinfection by SARS-CoV-2 true reinfections?"
we corrected the question as follows: "are all cases reported in the literature as reinfection by SARS-CoV-2 true reinfections?"

Lines 272/286: Replace the last comma with "and".
we replaced the last comma with "and".

Line 289: Replace the first comma with "and".
we replaced the first comma with "and".

Line 296: As viruses are not cultivable agents, change "virus culture" to "virus isolation".
we changed "virus culture" to "virus isolation".

Line 303: Remove the extra space after "lasting".
we removed the extra space after "lasting".

Line 304: Change "for" to "due to".
we changed "for" to "due to".

Line 308: Remove the lowercase "r" in "rRT-PCR".

We removed the lowercase "r" in "rRT-PCR".

Line 317: In Figure 2, remove the red underlines and change "viral culture" to "virus isolation".

In Figure 2, we removed the red underlines and change "viral culture" to "virus isolation".

Line 323: With "unwanted host", did the authors mean "unwanted virus"?

We correct in virus

Line 327: Remove the extra space before "However".

We removed the extra space before "However".

Line 330: Change "virus" to "virion".

We changed "virus" to "virion"

Line 339: Remove "continue to" before "remain".

We removed "continue to" before "remain".

Lines 352-353: Please consider that anti-SARS-CoV-2 IgM antibodies were already shown to persist for up to 8 months post-COVID-19 (DOI: 10.4236/crcm.2021.109029).
we modified reporting that anti-SARS-CoV-2 IgM antibodies were already shown to persist for up to 8 months post-COVID-19 (DOI: 10.4236/crcm.2021.109029).

Line 371: Replace the last comma with "and".
we replaced the last comma with "and".

Line 393: COVID-19 is the disease, not the etiological agent.

We substituted COVID-19 with SARS-CoV-2

Line 404: Put the first letter in "Anti-SARS-CoV-2" in lowercase.

We modified the first letter in "Anti-SARS-CoV-2" in lowercase

Line 414: Remove the extra space before "mutations".
we removed the extra space before "mutations".

Line 422: Replace the comma after "cytomegalovirus" with "and".

We replaced the comma after "cytomegalovirus" with "and".

Line 423: Reverse the word order in "diagnosis differential".

We reversed the word order in "diagnosis differential".

Line 436: Insert a hyphen between "antibody" and "dependent".

We Inserted a hyphen between "antibody" and "dependent".

Line 443: The antibody-dependent enhancement phenomenon had already been named in line 436.
we deleted the sentence.

Line 453: Put the second "s" in "with SARs-CoV-2" in upper case and insert a hyphen between "Cov" and "2" in "SARS-CoV2 RNA".

We modified the second "s" in "with SARs-CoV-2" in upper case and inserted a hyphen between "Cov" and "2" in "SARS-CoV2 RNA".

Line 480: Add "-19" after  "COVID" in "COVID episode".

We added "-19" after  "COVID" in "COVID episode".

Line 506: Add "by" after "reported".

We added "by" after "reported"

Line 507: Put "recognize" in the gerund.

We modified “recognize" in “recognizing”.

Line 530: Change "form" to "from" and remove the extra space after "literature".
we changed "form" to "from" and removed the extra space after "literature".

Line 542: Put "covid-19" in uppercase and remove "as seen sometimes" after "since".

We modified "covid-19" in uppercase and removed "as seen sometimes" after "since

Round 2

Reviewer 1 Report

There are some grammar errors and typos. Please revise them before publishing.

Reviewer 2 Report

The authors responded adequately to my comments on the original version by improving English language as well as data exposure, thus I recommend accepting the article for publication.